# Recurrent network dynamics shape direction selectivity in primary auditory cortex

Destinee A. Aponte[1,2], Gregory Handy [3,4,5], Amber M. Kline [1,2], Hiroaki Tsukano[1,2], Brent Doiron[3,4,5] & Hiroyuki K. Kato [1,2,6✉]

Detecting the direction of frequency modulation (FM) is essential for vocal communication in both animals and humans. Direction-selective firing of neurons in the primary auditory cortex (A1) has been classically attributed to temporal offsets between feedforward excitatory and inhibitory inputs. However, it remains unclear how cortical recurrent circuitry contributes to this computation. Here, we used two-photon calcium imaging and whole-cell recordings in awake mice to demonstrate that direction selectivity is not caused by temporal offsets between synaptic currents, but by an asymmetry in total synaptic charge between preferred and non-preferred directions. Inactivation of cortical somatostatin-expressing interneurons (SOM cells) reduced direction selectivity, revealing its cortical contribution. Our theoretical models showed that charge asymmetry arises due to broad spatial topography of SOM cell-mediated inhibition which regulates signal amplification in strongly recurrent circuitry. Together, our findings reveal a major contribution of recurrent network dynamics in shaping cortical tuning to behaviorally relevant complex sounds.

---

[1] Department of Psychiatry, University of North Carolina at Chapel Hill, Chapel Hill, NC 27599, USA. [2] Neuroscience Center, University of North Carolina at Chapel Hill, Chapel Hill, NC 27599, USA. [3] Departments of Neurobiology and Statistics, University of Chicago, Chicago, IL, USA. [4] Department of Mathematics, University of Pittsburgh, Pittsburgh, USA. [5] Grossman Center for Quantitative Biology and Human Behavior, University of Chicago, Chicago, IL, USA. [6] Carolina Institute for Developmental Disabilities, University of North Carolina at Chapel Hill, Chapel Hill, NC 27599, USA. ✉email: hiroyuki_kato@med.unc.edu

Direction selectivity represents a fundamental computation found across sensory modalities. For example, neurons in the visual cortex respond selectively to objects moving in one direction, and those in the somatosensory cortex prefer particular directions of whisker deflections. In the auditory system, neurons fire selectively to upward or downward frequency modulations (FM), which provides critical substrates for auditory-guided behaviors, such as echolocation and vocal communication[1–3]. Revealing how neural circuits enable direction-selective firing is a fundamental step toward understanding how our brain integrates information over time to interpret dynamic sensory inputs from the external world.

Decades of studies on direction selectivity in various sensory systems have given rise to two competing algorithms. One model proposes a delay line mechanism, where stimuli on neighboring parts of a receptive field trigger excitatory inputs with distinct latencies[4], while the second model proposes that non-preferred direction of movement evokes leading inhibition which suppresses spikes triggered by trailing excitation[5]. Despite their difference in synaptic mechanisms, these models are similar in that they both attribute direction selectivity to temporal offsets between feedforward synaptic inputs onto an integrating neuron. In contrast to these feedforward mechanisms for direction selectivity, surprisingly little is known regarding the roles of recurrent circuits, which constitutes around 95% of synapses in sensory cortices[6,7]. Studies in anesthetized mice have reported that cortical activity precisely follows thalamic inputs and linearly multiplies the signal, arguing against an active computational role of recurrent circuits in direction selectivity[8,9]. Accordingly, direction selectivity in both auditory and visual cortices of anesthetized animals has been attributed to either the feedforward mechanisms described above[10–15] or the inheritance from upstream subcortical structures[16–20].

Recently, a number of studies in awake animals have challenged the feedforward-dominant view of cortical operation by demonstrating that cortex operates as an inhibition-stabilized network (ISN)[21–27]. ISN is a circuit operation regime in which excitatory recurrence is strong enough to destabilize neural activity unless stabilized by feedback inhibition, giving rise to nonlinear, and even paradoxical, cortical activities due to network-level interactions[28–30]. These findings raise a question of whether recurrent cortical circuit simply follows thalamic inputs in the awake state, or has more active roles in shaping cortical tuning to complex sensory stimuli. Here, we combine two-photon calcium imaging, whole-cell recording, and computational modeling to dissect the circuit mechanisms underlying FM direction selectivity in the neurons of the primary auditory cortex (A1) in awake mice. In contrast to classical models, we found that direction selectivity is not caused by temporal offsets between feedforward synaptic inputs; rather, it is generated due to differential amplification of input signals in the recurrent circuitry between preferred and non-preferred directions. These results demonstrate that cortical tuning to temporally complex sensory stimuli are shaped by nonlinear recurrent network dynamics, a conclusion that moves away from the classical idea of feedforward-dominant circuitry.

## Results

**A1 neurons are direction-selective to sweeps with ethologically relevant FM rates.** To determine the ethological range of FM rates used in mouse vocalizations, we first conducted recordings of their communication calls. We focused on three categories of vocalizations—pup isolation calls, male courtship songs, and pain vocalizations, which are known to have distinct spectro-temporal structures (Fig. 1a, Supplementary Fig. 1)[31–33]. We collected

vocalizations from C57BL/6J, BALB/c, and CBA mice since vocalization patterns vary across strains. We found that the absolute FM rate rarely exceeds 40 oct/s in all three categories of vocalizations (Fig. 1b; pup call: 2.3%; male song: 1.7%; pain vocalization: 1.4%). Average absolute FM rates were 9.9 oct/s for pup call, 11.0 oct/s for male song, and 7.5 oct/s for pain vocalization. These data show that mice mostly use slower ranges of FM rates for communication than what has been tested in many of previous studies (30–90 oct/s[14], 70 oct/s[8,15], or 8–670 oct/s[34]), which led us to focus our experiments on these ethologically meaningful FM rates.

We next asked how directions of slow FM sweeps are encoded in A1 neurons by conducting two-photon calcium imaging in awake head-fixed mice (Fig. 1c). To identify A1 location, tonotopy was mapped with intrinsic signal imaging through the skull using pure tones of three frequencies (3, 10, and 30 kHz). We injected adeno-associated virus (AAV) vectors to express calcium indicator GCaMP6s in A1 of transgenic mice in which GABAergic interneurons are marked with tdTomato (*Vgat-IRES-Cre* × *ROSA-LSL-tdTomato*). This allowed us to optically distinguish glutamatergic pyramidal cells (green) from GABAergic interneurons (green + red). Two to three weeks following virus injection and implantation of a glass window over A1, we performed two-photon calcium imaging in layer 2/3 (L2/3) to measure responses of individual neurons to FM sweeps. Tuning properties of L2/3 pyramidal neurons were determined by presenting upward or downward sweeps of various rates (2.5–80 oct/s, 6 rates in each direction). To evoke responses in A1 neurons with a wide range of frequency preference, long FM sweeps with 4-octave range (4–64 kHz) were presented at 70 dB SPL. Overall, 33% of GCaMP6s-expressing pyramidal cells ($n =$ 429 out of 1292 cells, 8 mice) increased their activity (measured as dF/F) in response to at least one sweep stimulus, and 68% of sweep-responsive neurons showed absolute direction selectivity index (DSI) values larger than 0.3. Importantly, we observed strong direction selectivity in many neurons even at slow FM rates (Fig. 1d), which is in contrast to previous studies in anesthetized rodents which reported the lack of direction selectivity for sweeps below 30 oct/s[15] or 0.5 kHz/ms[35]. The fraction of responsive neurons monotonically increased from fast to slow FM sweeps, likely reflecting the larger amount of total sound energy transmitted by slow sweeps (Fig. 1e). Nevertheless, the absolute DSI value remained high regardless of FM rate (Fig. 1f); we also confirmed this result with single-unit recordings and found enhanced direction selectivity for slow FM sweeps in the awake state compared to the anesthetized state (Supplementary Fig. 2). When DSI was compared against the best frequency (BF) of individual neurons, we found a negative correlation between DSI and BF in both pyramidal (Fig. 1g, h) and GABAergic neurons (Supplementary Fig. 3), with higher absolute DSI values at the edges of A1 tonotopy. We also note a significant local heterogeneity in DSI values, which did not stringently follow the global trend. This heterogeneity became less prominent as we raised the data selection threshold (Supplementary Fig. 4), indicating that cells with less robust responses showed larger variability in DSI. This selection criteria-dependence of local heterogeneity is reminiscent of that in A1 tonotopy[36], and thus it might represent a general rule of spatial organization in this area. Together, these results demonstrate that A1 neurons in awake state discriminate between directions of FM sweeps at ethologically relevant rates.

**Direction selectivity is caused by asymmetry of excitatory postsynaptic charge between FM directions.** To directly determine postsynaptic currents that shape direction selectivity, we

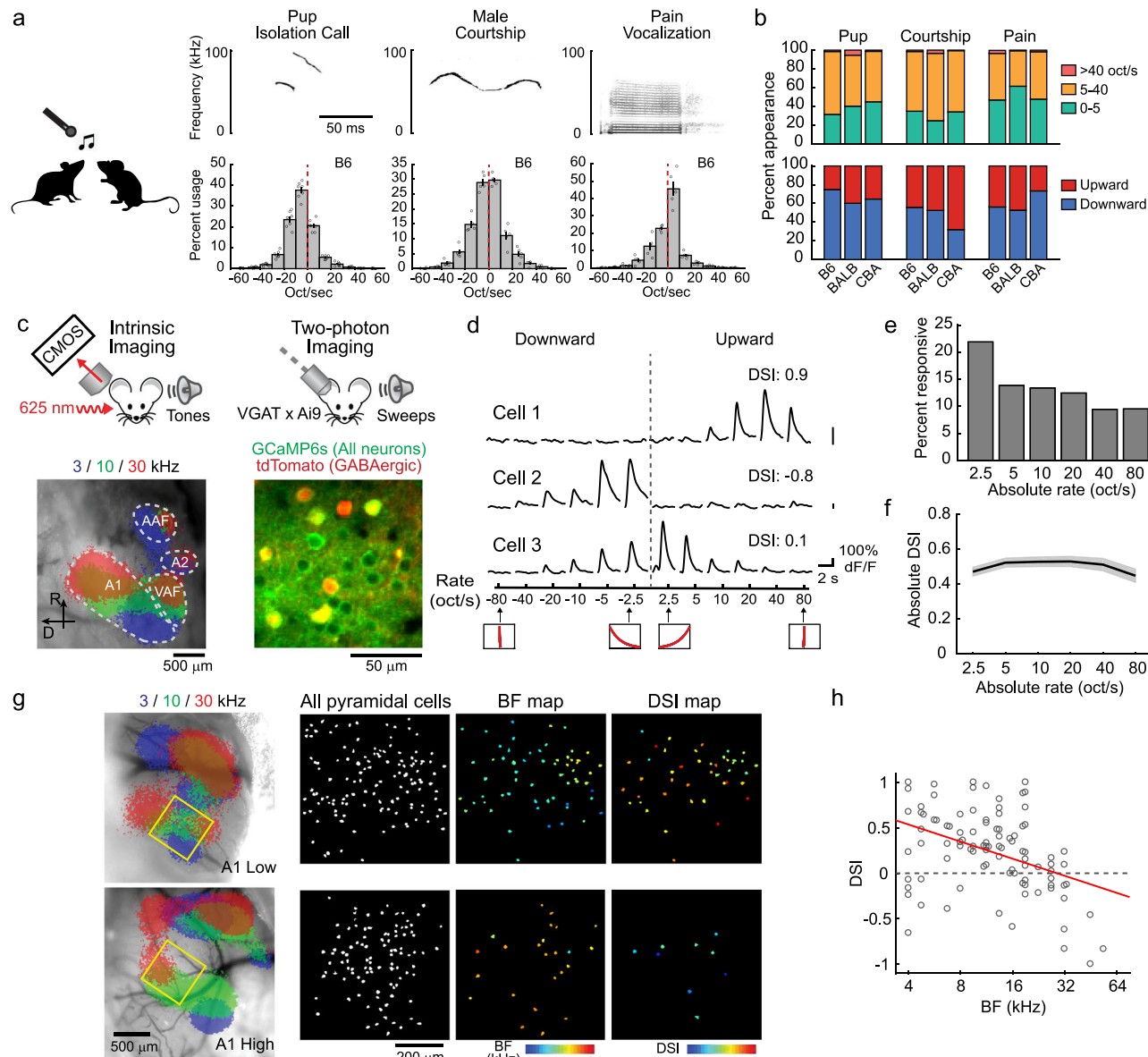

**Fig. 1 A1 neurons are direction-selective to sweeps at ethologically relevant FM rates. a** Left, vocalization recording schematic. Top right, spectrograms of representative vocalizations. Bottom right, histograms showing the usage probability of FM rates for three vocalization categories in C57BL/6J mice, overlaid with individual data points ($n = 7, 5, 5$ mice, 20,295, 14,719, 17,826 vocal contour fragments for pup, male, and pain vocalizations; see Supplementary Data 1). Results are mean ± SEM. **b** Top, usage probability of FM rates for three vocalization categories in three strains. Bottom, usage probability of upward and downward sweeps (BALB: $n = 8, 3, 3$ mice, 4164, 13,671, 12,235 vocal contour fragments; CBA: $n = 4, 3, 3$ mice, 21,658, 11,314, 25,640 vocal contour fragments for pup, male, and pain vocalizations). **c** Left, intrinsic signal imaging of responses to pure tones superimposed on cortical surface imaged through the skull. Right, in vivo two-photon image of L2/3 neurons in A1. A1 tonotopy was reproducibly observed in all eight mice. **d** FM sweep tuning of three representative L2/3 pyramidal cells. Traces are average responses (five trials). Insets at the bottom show the schematics of frequency versus time representations. DSI was calculated as $(U - D)/(U + D)$, where U and D represent the responses triggered by upward and downward sweeps, respectively. **e** Fraction of responsive cells at six absolute FM rates. ($n = 8$ mice, 1292 cells). **f** Average (solid line) and SEM (shading) of absolute DSI at each FM rate ($n = 8$ mice, 205 sweep-responsive cells). **g** Cellular-level spatial organization of BF and DSI in two representative A1 areas. Left, intrinsic signal image superimposed on cortical vasculature imaged through a glass window. Yellow squares represent the two-photon imaging fields of view. Right, maps showing the location of imaged pyramidal cells, BF for pure tones, and DSI for FM sweeps. **h** DSI of pyramidal cells averaged across all FM rates have a strong dependence on their BF ($n = 8$ mice, 96 cells responsive to both sweeps and pure tones. $R = -0.403$, $p = 4.6 \times 10^{-5}$, two-sided *t*-test). Red line, regression curve.

next performed whole-cell patch-clamp recordings in awake mice (Fig. 2a). We targeted recording pipettes to L2/3 of A1, guided by the tonotopy determined by intrinsic signal imaging. Consistent with our previous study[22], recordings in awake mice revealed sustained, high-frequency barrages of spontaneous EPSCs and IPSCs, indicating that cortical circuits were highly active even in the absence of delivered sounds. Interestingly, presentation of slow FM sweeps consistently triggered a long-lasting suppression of both spontaneous EPSCs and IPSCs, which has been previously termed "network suppression"[22,37] (marked by asterisks in Fig. 2b, c; on trial-averaged traces, network suppression appears as slow currents which drop below baseline). Network

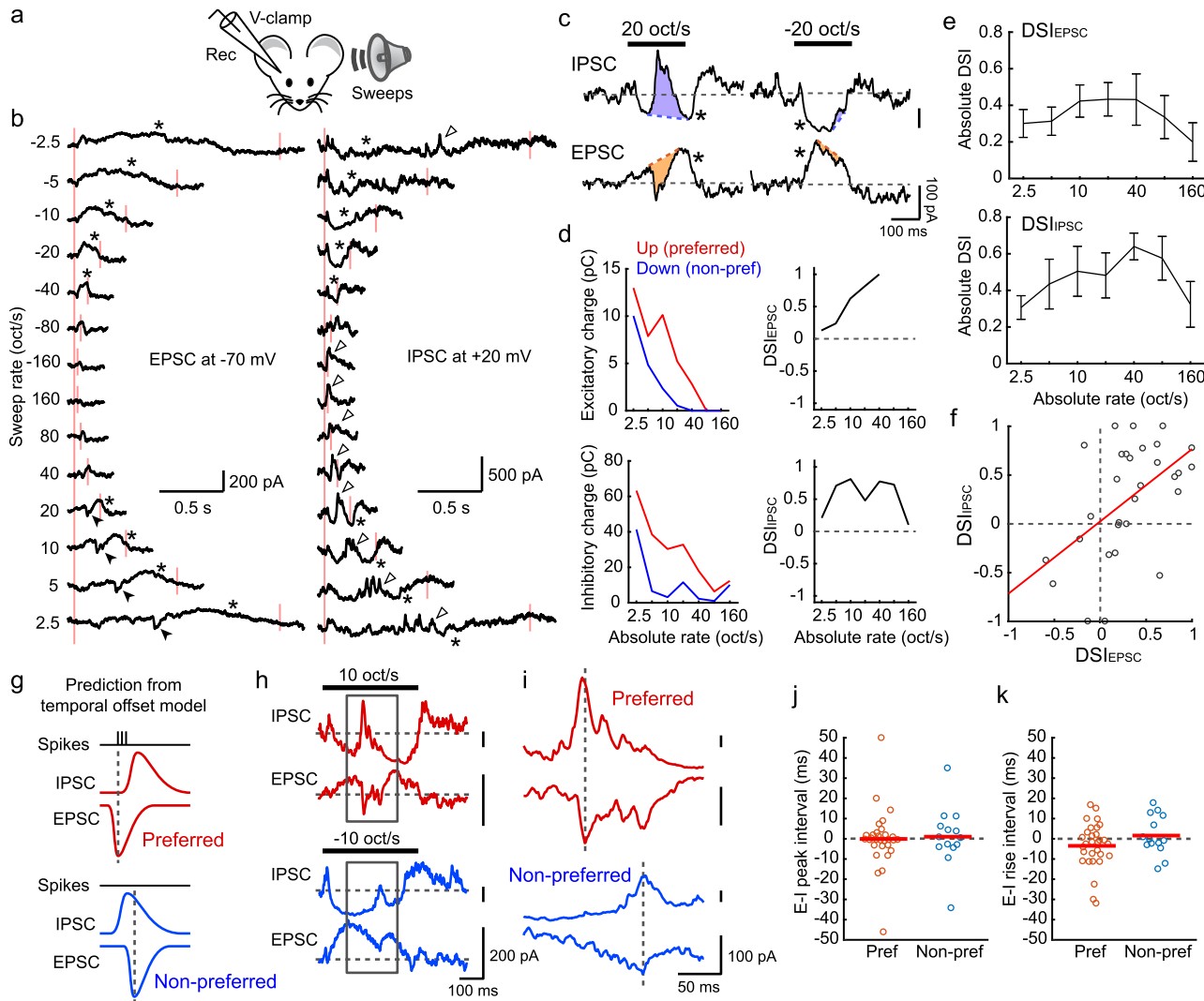

**Fig. 2 Direction selectivity is caused by the asymmetry of sound-evoked postsynaptic charge between FM directions. a** In vivo whole-cell recording schematic. **b** EPSCs (left) and IPSCs (right) evoked by FM sweeps with different rates. Traces are an average of six trials. Red lines, sound onsets and offsets. Black arrowheads, fast EPSCs. Open arrowheads, fast IPSCs. Asterisks, network suppression. **c** Magnified EPSCs and IPSCs from (**b**) at +20 oct/s (left) and −20 oct/s (right) FM sweeps. Orange (EPSC) and purple (IPSC) shaded areas highlight sound-evoked fast postsynaptic currents which are prominent only in preferred direction. **d** Quantitative analysis for the cell shown in (**b**) at each absolute FM rate. Left, the total charge of fast postsynaptic currents. Right, $DSI_{EPSC}$ and $DSI_{IPSC}$ calculated from the charge. No data point is shown for rates that did not trigger postsynaptic currents. **e** $DSI_{EPSC}$ and $DSI_{IPSC}$ averaged across cells (EPSC, $n = 5$ mice, 9 cells; IPSC, $n = 4$ mice, 7 cells). Results are mean ± SEM. **f** $DSI_{EPSC}$ and $DSI_{IPSC}$ are correlated ($R = 0.522$; $p = 0.0031$, two-sided $t$-test; $n = 30$ cell-rate pairs with DSI values for both EPSCs and IPSCs). Red line, regression curve. **g** Predicted timings of EPSCs, IPSCs, and spikes triggered by preferred and non-preferred FM sweeps in the temporal offset model. Dotted lines, peak timings of EPSCs. **h** EPSCs and IPSCs evoked by preferred (10 oct/s; top) and non-preferred (−10 oct/s; bottom) FM sweeps in another representative cell. **i** Magnified view of traces within gray boxes in (**h**). **j** Time by which EPSC peak precedes IPSC peak is not significantly different from zero in either direction (preferred: $n = 30$ cell-rate pairs, $p = 0.976$; non-preferred: $n = 14$, $p = 0.800$; preferred vs non-preferred: $p = 0.817$, two-sided $t$-test). Red lines indicate mean. **k** The time by which EPSC onset precedes IPSC onset is not significantly different from zero in either direction (preferred: $p = 0.100$; non-preferred: $p = 0.549$; preferred vs non-preferred: $p = 0.153$, two-sided $t$-test).

suppression is a suppression of recurrent activity, which is evoked by a broad range of non-preferred frequencies; therefore, sustained network suppression is triggered as our 4-octaves FM sweeps pass through these non-preferred frequencies. In addition to network suppression, preferred FM sweeps evoked fast EPSCs (filled arrowheads in Fig. 2b and orange shaded areas in Fig. 2c) as expected from their crossing of the neuron's tonal receptive field (TRF). Remarkably, non-preferred FM sweeps triggered smaller and often undetectable fast EPSCs, despite their crossing of the TRF (Fig. 2b–d). This asymmetry in charge between directions could be a source of direction selectivity in neuronal firing. To quantify the attenuation of synaptic currents in the

non-preferred direction, we calculated DSI values for excitatory and inhibitory postsynaptic charges ($DSI_{EPSC}$ and $DSI_{IPSC}$). Absolute $DSI_{EPSC}$ was high across the entire range of sweep rates and was consistent with that of neuronal firing (Fig. 2e; see Fig. 1f). Similar to fast EPSCs, we observed a strong attenuation of fast IPSCs (open arrowheads in Fig. 2b and purple shaded areas in Fig. 2c) in non-preferred FM sweeps, and $DSI_{IPSC}$ followed the same trend as $DSI_{EPSC}$ (Fig. 2e, f). We observed a strong dependence of $DSI_{EPSC}$ and $DSI_{IPSC}$ on the BF of individual neurons (Supplementary Fig. 5), further demonstrating the parallel between direction selectivity of neuronal firing and postsynaptic charges (Fig. 1h).

To directly examine a previous model that proposed temporal offsets between EPSCs and IPSCs (Fig. 2g), we next quantified the relative timing of EPSCs and IPSCs during FM sweeps (Fig. 2h–k). Notably, we did not observe a difference in EPSC-IPSC time intervals between preferred and non-preferred directions, regardless of whether we measured peak timing or onset timing (Preferred vs Non-preferred, peak: $p = 0.817$; onset: $p = 0.153$; Fig. 2j, k). EPSC-IPSC time intervals clustered around zero for both measures, indicating the covariance of excitation and inhibition. In fact, opposite from the temporal offsets hypothesis, the onset timing of IPSCs tended to lead that of EPSCs in the preferred direction ($-3.5 \pm 2.0$ ms, $p = 0.100$), likely reflecting the slightly broader frequency tuning of IPSCs than EPSCs[22,38]. Together, these findings argue against temporal offsets between feedforward synaptic inputs as the source of direction selectivity in A1; our results rather indicate that in awake mice, direction selectivity is determined by an asymmetry in total postsynaptic charge between sweep directions.

**SOM cells shape direction selectivity in A1.** What are the circuit mechanisms underlying the attenuated excitatory post-synaptic inputs in non-preferred directions? Previous studies in anesthetized animals proposed that the cortex linearly multiplies thalamic inputs and that the direction selectivity existing in postsynaptic charges is simply inherited from the thalamus[8,15]. Alternatively, we considered the possibility that a strongly recurrent awake cortex[25] further shapes direction selectivity in A1. Specifically, we focused on the role of network suppression—suppression of recurrent activity triggered by non-preferred tones—because its slow kinetics and asymmetry within TRFs[22] make it suited for the directional interaction between frequency channels (see below). Since network suppression requires SOM cells, but not PV cells[22], we inactivated these two interneuron subtypes to examine their effects on direction selectivity. We performed single-unit recording in A1 of head-fixed awake mice and photoinactivated SOM (Fig. 3a–c) or PV (Fig. 3d–f) cells using virally introduced halorhodopsin (eNpHR3.0). We used weak LED intensities to modulate transmission without inducing irreversible changes in the cortical activity states (see "Methods"). Remarkably, SOM cell photoinactivation that had little effect on responses to preferred FM sweeps significantly increased the responses of regular-spiking units to non-preferred FM sweeps (Fig. 3b, c and Supplementary Fig. 6a). As a result, absolute DSI values were significantly reduced in LED trials compared to the interleaved control trials ($p = 0.0006$; Fig. 3g; see Supplementary Fig. 7a for fast-spiking units). This result supports the role of SOM cells in shaping direction selectivity within the auditory cortex.

In contrast to the SOM inactivation experiments, we did not observe any decrease in absolute DSI during PV cell photo-inactivation (Fig. 3d–g and Supplementary Fig. 6b). The absolute DSI rather showed a tendency for a slight enhancement, which could be explained by previously reported enhanced network suppression during PV cell photoinactivation[22]. The difference between SOM and PV photoinactivation effects cannot be attributed to different levels of optogenetic manipulation, since the increase in the spontaneous firing was indistinguishable between SOM and PV cell photoinactivation ($p = 0.669$; Fig. 3h and Supplementary Fig. 7b). Together, these results rule out the possibility that direction selectivity in A1 is simply inherited from subcortical systems and demonstrate active shaping of cortical direction selectivity by SOM inhibitory neurons.

**ISN model explains the nonlinear integration of FM sweeps in the A1 circuit.** Our data indicate that the asymmetry in excitatory postsynaptic charge between FM directions is enhanced within A1, suggesting a nonlinear amplification of thalamic inputs in cortical circuitry. What circuit properties enable this computation, and what is the reason for the difference between our results and previous reports in anesthetized animals? In order to examine these central questions, we constructed a network model of A1 to fully explore the circuit and cellular parameters that are not accessible by experimental manipulations. To this end, we built a three-population network model that incorporated known connectivity parameters across neuronal subtypes (pyramidal, PV, and SOM) as well as the spatial constraints within A1 tonotopy (Fig. 4a and "Methods")[28,39]. Motivated by past experimental recordings in A1 from awake mice[22,24] we modeled the network to be an ISN[28–30]. To simplify our analysis, we modeled individual neuronal activity with a dynamic firing rate and organized neurons along a one-dimensional spatial continuum of tonotopy (Fig. 4b). Although this simplification ignores the local heterogeneity in DSI distribution (Fig. 1h), it aims to capture the global trend for the dependence of DSI on the preferred frequency of individual neurons. A threshold-linear firing rate function was used to prevent the firing rates from going below zero (Fig. 4c). Spatially broad effective connectivity was taken for both inputs and outputs of SOM cells, while all other connections were set to be narrow[22]. We modeled an upward (downward) FM sweep as a traveling Gaussian pulse that begins at the low (high) frequency position in the tonotopy (Supplementary Movie 1). We assume that DSI of the thalamic input is zero (see "Discussion"). In total, this feedforward thalamic input activates the recurrent network, which both temporally filters and spatially convolves the activity to give the final cortical response.

With this model in hand, we start by considering how a 20 oct/s upward FM sweep drives excitatory neurons with a BF of 5 kHz (Fig. 4d, red curves). While the FM sweep is near this BF, these neurons receive direct feedforward excitation, which is amplified via recurrent excitatory connections (Fig. 4di-dii). At the same time, they receive strong inhibition from nearby PV cells, as these interneurons also receive feedforward and recurrent excitation (Fig. 4diii). Inhibition from SOM cells, on the other hand, peaks when the FM sweep is exciting neurons with a higher BF, due to their spatially broad connectivity and slow kinetics (Fig. 4div). This delayed input from SOM cells suppresses recurrent activity and brings both pyramidal and PV cells below their baseline firing (network suppression). The combination of all these feedforward and recurrent inputs produce an initial strong increase in firing rates, followed by network suppression (Fig. 4dv). In contrast, considering a $-20$ oct/s downward FM sweep, the narrative plays out in a similar fashion, except that the pyramidal cells experience leading network suppression from SOM cells before the arrival of feedforward excitation (Fig. 4d, blue curves and Supplementary Fig. 8). This network suppression silences the recurrent activity and leads to an attenuated amplification of both excitatory and inhibitory inputs (Fig. 4e and Supplementary Fig. 9a). Therefore, this differential amplification between upward and downward directions results in a positive DSI (Fig. 4f).

Is the threshold nonlinearity we introduced in the model, as opposed to a completely linear model, required to capture direction selectivity? This influence of the threshold can easily be removed by increasing the baseline firing rate of excitatory neurons, leading to a completely linear network model. Interestingly, if this is the case, we can prove that DSI = 0 for all choices of model parameters (see Supplementary Text). This important insight shows that, in addition to modeling the proper circuit structure, we must also consider a nonlinear neuronal transfer function in order to explore the mechanistic underpinnings of direction selectivity.

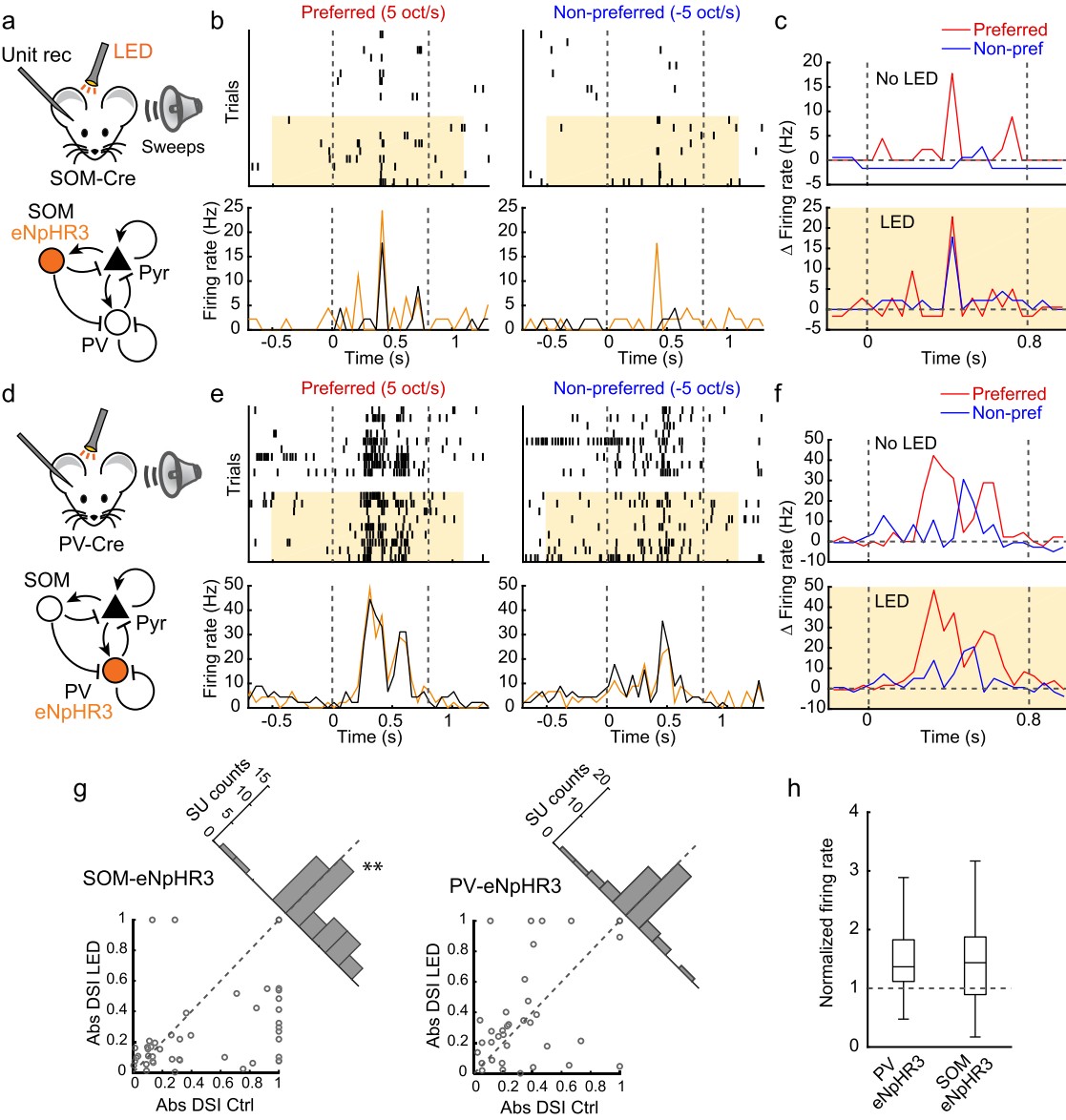

**Fig. 3 SOM cells shape direction selectivity in A1. a** Schematics of optogenetic inactivation of SOM cells during single-unit recording. **b** FM sweep responses with (amber shades and traces) and without (black traces) SOM cell photoinactivation. Top, raster plots of FM sweep responses in a representative regular-spiking single-unit. Bottom, peristimulus time histogram (PSTH). Control and photostimulation trials were interleaved but are separated here for clarity. Gray dotted lines indicate sound onsets and offsets. **c** PSTHs in (**b**) shown separately for control (top) and photostimulation (bottom) trials after subtracting the baseline firing rate just before sounds. **d** Schematics of optogenetic inactivation of PV cells during single-unit recording. **e–f** FM sweep responses with and without PV cell photoinactivation. **g** Left, scatter plot showing absolute DSI during control and SOM cell photoinactivation trials. Gray dots show all sweep-responsive single-units ($n = 4$ mice, 48 out of 111 regular-spiking single-units). The oblique histogram illustrates the changes in absolute DSI with LED (**$p = 0.0006$, two-sided $t$-test). Right, the same plots for PV cell photoinactivation. $n = 4$ mice, 45 out of 99 regular-spiking single-units. $p = 0.667$. Dotted lines, unity lines. **h** Photoinactivation of PV cells and SOM cells increased the spontaneous firing rate to a similar degree ($n = 88$ and 96 regular-spiking units with baseline firing rate > 0.25 Hz; $p = 0.669$, two-sided $t$-test). Box plots show median and 25th and 75th percentiles as box edges, and 1.5 × interquartile range as whiskers.

Since our model is organized along the tonotopy, the circuit is symmetric about the BF of 16 kHz. For all neurons with a BF < 16 kHz, the time courses are similar to the one shown for neurons with 5 kHz BF and DSI will be positive, although the magnitude depends on their exact location. For all neurons with BF > 16 kHz, the upward and downward time courses are essentially interchanged (e.g., leading suppression occurs for upward sweeps) and DSI will be negative (Fig. 4f). We note that this result and the qualitative characteristics of these curves hold true for a wide range of FM rates, providing evidence that our model explains direction selectivity for FM

sweeps at ethologically relevant rates (Fig. 4g and Supplementary Fig. 8).

We next tease apart this model and identify the crucial features contributing to the shape of the DSI curve. We start by modeling the photoinactivation of SOM and PV cells (Fig. 3) by providing a hyperpolarizing current to these interneurons and weakening the effective connectivity. Inactivating SOM cells results in a major reduction in absolute DSI across the tonotopic axis (Fig. 5a). While leading suppression is still present, it is significantly weaker and insufficient to silence recurrent activity. As a result, excitatory inputs are amplified in a similar amount during upward and

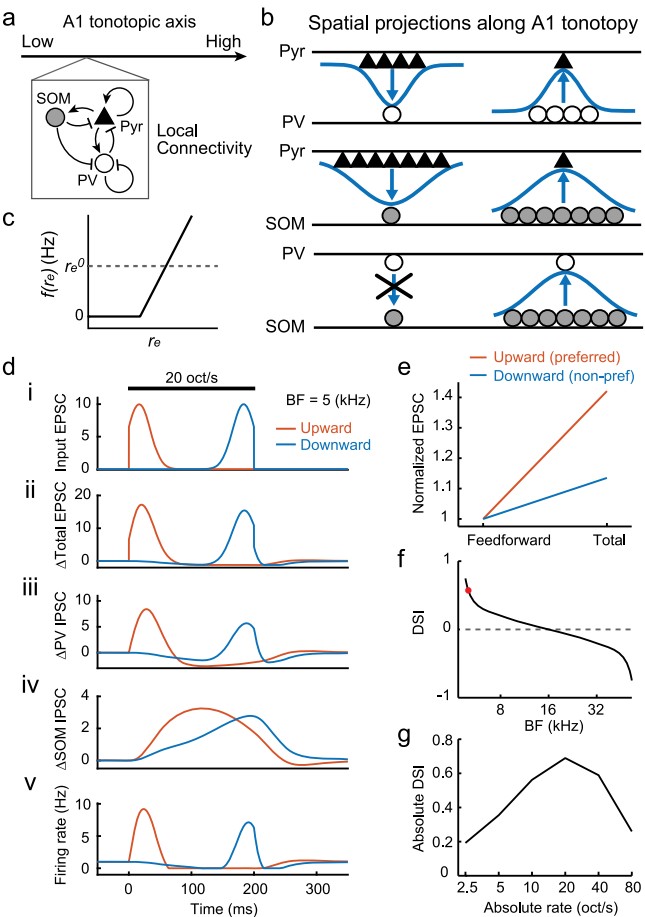

**Fig. 4 ISN model explains the nonlinear integration of FM sweeps in the A1 circuit. a** Schematics of local connectivity between three cell populations. **b** Schematics showing spatial scales of connectivity between populations. Horizontal lines show a one-dimensional spatial continuum of tonotopy. **c** Neuronal transfer function from input current to output firing rate. **d** Time courses of feedforward EPSC (i), change in total EPSC (ii), change in IPSC from PV cells (iii), change in IPSC from SOM cells (iv), and firing rate (v) of a pyramidal cell with 5 kHz BF. Dark line represents FM sweep stimulus at ±20 oct/s. **e** Summary plot showing a differential amplification of excitatory postsynaptic charges between upward and downward directions. Amplitudes are normalized to that of feedforward EPSC. **f** DSI plotted against BF at ±20 oct/s FM rates. Data point with BF = 5 kHz is shown as a red dot. **g** Absolute DSI plotted against absolute FM rates.

downward sweeps. In contrast, inactivating PV cells has the opposite effect, amplifying the leading network suppression[22], prolonging the period of silence, and maintaining, if not strengthening, DSI (Fig. 5b). Thus, our model recapitulates the critical role of SOM cells in generating direction selectivity (Fig. 3). Interestingly, solely reducing the spatial integration and projection scales of SOM cells to match those of pyramidal and PV cells also eliminates the leading network suppression and, as a result, direction selectivity (Fig. 5c). This result emphasizes the importance of SOM cells' broad connectivity in linking distal frequency domains of A1 circuits.

Lastly, we investigate how a shift from an ISN to non-ISN regime, a change potentially resembling a shift from awake to anesthetized states (see "Discussion"), affects these results. This is done by weakening the effective recurrent interactions between pyramidal neurons and decreasing the spontaneous firing rate to 0.1 Hz. We find that the weakening of recurrent connectivity is enough to push the system to operate in a linear regime and eliminate both network suppression and direction selectivity (Fig. 5d). While it would be tempting to thus conclude that ISN dynamics are a necessary condition for direction selectivity, this is not yet evident. Our result illustrates that the circuit is more likely to reach the non-linearity threshold when it has strong recurrent interactions that allow a large dynamic range of firing rates

through signal amplification. Although building a robust circuit with strong recurrence likely calls for an ISN regime[29,30], it is possible that a carefully tuned non-ISN network could reach the nonlinearity threshold, albeit over a restricted range of stimulus parameters.

Taken together, our simulation results provide a theoretical foundation for nonlinear amplification of inputs in A1 and show three circuit components that contribute to the generation of FM direction selectivity: a nonlinear neuronal transfer function, spatially broad connectivity of SOM cells, and strong recurrent connectivity.

**A1 neurons are direction-selective to FM sweeps with restricted frequency ranges.** Our results illustrate cortical circuit mechanisms that shape direction selectivity for ethologically relevant slow FM sweeps. Could similar mechanisms generate direction selectivity to spectrally-restricted FM sweeps which are prevalent in mouse vocalizations (Supplementary Fig. 1c)? We experimentally tested this by imaging GCaMP6s-expressing L2/3 pyramidal cells while presenting FM sweeps that covered one octave of the frequency range (lowest range: 4–8 kHz; highest range: 32–64 kHz; Fig. 6a, b). Interestingly, we found that many neurons reversed their direction preference depending on the frequency range of the FM sweeps (Fig. 6b). This result indicated that FM direction

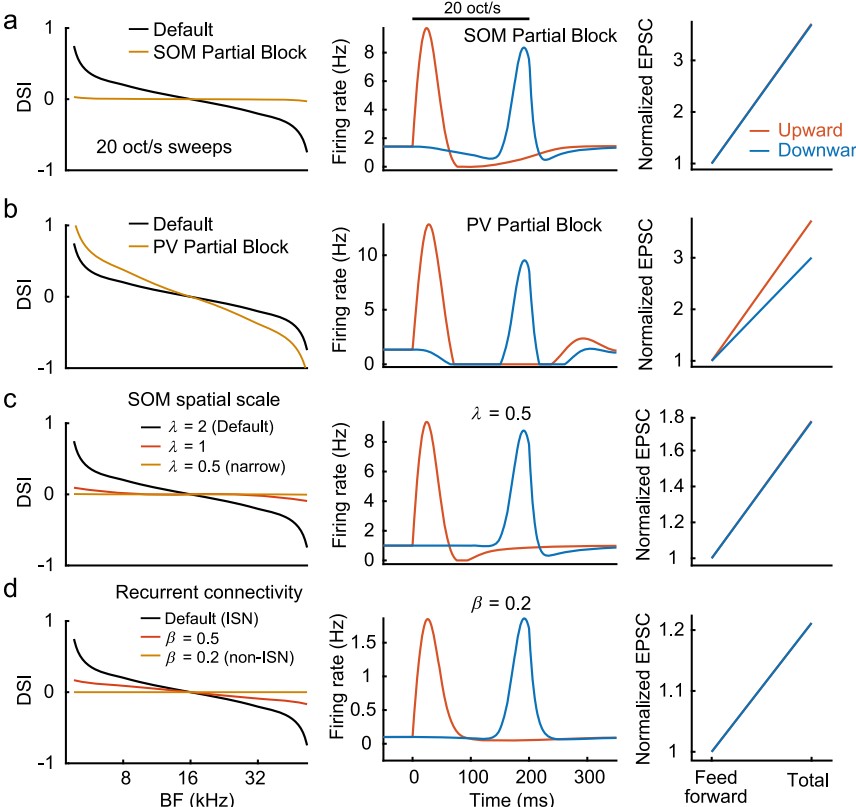

**Fig. 5 Network model demonstrates the requirement for SOM cell activity and strong recurrent excitation. a** Left, DSI plotted against BF at ±20 oct/s FM rates in the control (black) and SOM cell partial inactivation (yellow) conditions. Middle, firing rate of a pyramidal cell (5 kHz BF) during SOM cell partial inactivation. Right, amplification of excitatory postsynaptic charges in response to upward and downward FM directions. **b** Results for PV cell partial inactivation. **c** Results for reduced spatial scales of SOM cell connectivity. **d** Results for non-ISN models with reduced recurrent excitation strengths.

selectivity is not a fixed property in each neuron, but rather it is determined by the relationship between its BF and the sweep frequency range. To systematically examine this relationship, we determined the logarithmic center of the sweeps ($F_{cent}$) and calculated the relative position of $F_{cent}$ compared to the BF of each neuron (see an inset in Fig. 6b). This generated seven cell-$F_{cent}$ pairs for each cell, and DSI was calculated separately for each of these pairs. As expected, pyramidal cells were most responsive when their BFs were within the sweep frequency range (relative $F_{cent}$ between −0.5 and 0.5; Fig. 6c). Absolute DSI value remained high regardless of FM rates, showing that A1 pyramidal cells are direction-selective to slow and spectrally-restricted ethologically relevant FM sweeps (Fig. 6d). Remarkably, when DSI was calculated separately for bins of relative $F_{cent}$ position, we found a sharp reversal of DSI at around relative $F_{cent} = 0$ (Fig. 6e), a result also captured in our computational model (Supplementary Fig. 9c). Therefore, a neuron shows preference to the direction in which FM sweeps move away from its BF. This is consistent with the results that we obtained with long 4-octave sweeps (Fig. 1h) and presents a more general description of the relationship between a neuron's FM direction selectivity and TRF. These data further strengthen our conclusion that A1 neurons discriminate the directions of ethologically realistic FM sweeps that are present in vocal communications.

## Discussion

### Nonlinear computation in an ISN. An increasing number of studies in awake animals have found that the sensory cortex operates as an ISN[21–27]. We previously found that, as a consequence of ISN operation in A1, non-preferred pure tones evoke network suppression, an attenuation of recurrent activity which

leads to suppression of both spontaneous EPSCs and IPSCs[22,37]. In this study, we further demonstrate that network suppression in ISNs plays a critical role in generating direction selectivity to ethologically relevant slow FM sweeps. A simple schematic (Fig. 7) provides an interpretation of our model regarding how network suppression attenuates EPSCs for non-preferred FM sweeps. As previously reported, network suppression is asymmetrically distributed within TRFs (Fig. 7a, b)[22]. We now model FM sweeps as sequential presentations of pure tone frequencies in ascending (upward) or descending (downward) order (Fig. 7c). Time-staggered summation of EPSCs results in a network suppression either leading or following an excitation, depending on the direction of the sweeps (Fig. 7d). Due to its slow kinetics, lagging network suppression does not affect leading excitation (Fig. 7d, left). In contrast, leading network suppression overlaps with lagging excitation and attenuates recurrent activity (Fig. 7d, right). Since cortical recurrent circuit amplifies feedforward thalamic inputs by 2–5 fold[8,9,40], attenuation of recurrent activity strongly reduces sound-evoked EPSCs. This blockade of amplification in the non-preferred direction generates charge asymmetry between directions. This model successfully explains BF-, rate-, and frequency range-dependence of DSI (Supplementary Fig. 10). Although sequential presentations of pure tones are experimentally challenging due to the fast dynamics of sounds, our network simulations (Figs. 4, 5) capture these mechanisms and identify critical circuit components. Importantly, our model clarifies that direction selectivity for ethologically relevant slow FM sweeps is enhanced due to the long-lasting kinetics of network suppression. This contrasts with classical models which considered only fast feedforward inputs. Our results suggest the importance of considering the nonlinear dynamics of recurrent

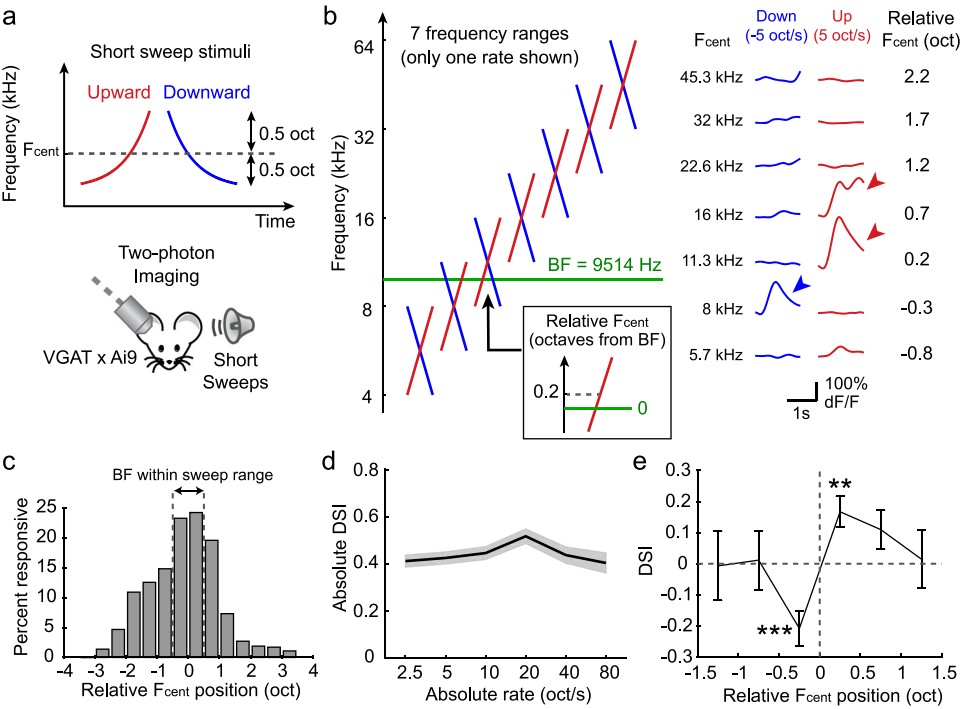

**Fig. 6 A1 Neurons are Direction-Selective to FM Sweeps with restricted frequency ranges. a** Schematic showing spectrally restricted FM sweeps with 1-octave frequency range. $F_{cent}$, logarithmic center. **b** Left, schematic showing seven frequency ranges used. Green line indicates the BF of a representative cell whose FM sweep responses are shown on the right. Inset shows the conversion of $F_{cent}$ to a relative position centered at the BF. Right, responses of a representative L2/3 pyramidal cell (BF = 9514 Hz) to FM sweeps with seven frequency ranges. This cell reversed its direction selectivity from upward preference (red arrowheads) to downward preference (blue arrowhead) around $F_{cent}$ at its BF. **c** The fraction of FM sweep responsive cells for each bin of relative $F_{cent}$ positions ($n = 6$ mice, 140 tone-responsive cells out of 485 imaged pyramidal cells). **d** Average (solid line) and SEM (shading) of absolute DSI at each FM rate (226 sweep-responsive cells). Only FM sweeps with $F_{cent}$ within ±1 octave from BF were included. **e** Averaged DSI plotted for each bin of relative $F_{cent}$ position, showing the reversal of DSI sign ($n = 25, 39, 58, 69, 60, 31$ responsive pyramidal cells at each $F_{cent}$). Results are mean ± SEM. ***$p = 0.0006$, **$p = 0.0012$, two-sided $t$-test.

circuits, which extends the ability of individual neurons to integrate information over time.

Many spatially distributed cortical models consider neurons distributed over a periodic domain, motivated by orientation selectivity in the visual system[41,42]. This provides a network symmetry where all neurons have inputs distributed over the same spatial scales. In our model, tonotopic organization is not on a periodic domain, and the spatial scale varies across the tonotopic location—neurons with low (high) BFs have inputs with shorter spatial scales on the low (high) frequency side of the tonotopic axis. As a result of this break in symmetry, all neurons receive inputs with distinct spatial footprints over the tonotopy. A sweeping input then drives unique patterns of spatio-temporal integration and imparts distinct direction selectivity to individual neurons. Therefore, it is essential to consider appropriate spatial constraints in modeling the cortical responses to complex stimuli.

**Direction selectivity in the primary auditory cortex.** Although previous studies have found a correlation between FM direction selectivity and the asymmetry of "sideband inhibition" within TRFs[43–45], synaptic mechanisms underlying this relationship remained controversial. We now demonstrate that network suppression, which accounts for the sideband inhibition in awake A1[22], generates FM direction selectivity not through temporal offsets between feedforward inputs, but through the attenuation of recurrent circuit-mediated amplification of excitatory inputs. Our results are consistent not only with the contribution of sideband inhibition, but also with previous findings on the roles of inhibitory neurons, including the reduction of direction selectivity with gabazine infusion[46], the

lack of effects on direction selectivity by PV cell photo-inactivation[12], and the role of SOM cells in triggering network suppression[22]. Thus, our proposed model resolves the previous controversy and provides an updated circuit basis for how sideband inhibition shapes direction selectivity in a recurrent network. However, we do not exclude the possibility that our model works in combination with other mechanisms that generate direction selectivity. For example, we observed a relatively low direction selectivity at extremely low (2.5 oct/s) and high (80–160 oct/s) FM rates in both experimental measurements of postsynaptic charges as well as model-based prediction, and these results deviate from the more flat relationship between DSI of neuronal firing and FM rates. We also observed local heterogeneity in DSI especially in cells with weak sweep responses, which cannot be explained by our one-dimensional A1 model. Direction selectivity at these conditions could be partially attributed to other mechanisms, such as delay-and-compare between ON and OFF responses[12], combination-sensitive supralinear summation[47], intensity ramp-selective firing[48] (see Supplementary text), and inheritance from upstream structures (see below).

Our SOM cell photoinactivation experiments demonstrated the enhancement of direction selectivity to slow FM sweeps within the cortex. In contrast, previous studies observed direction selectivity only at extremely fast FM rates (around 70 oct/s or 0.5–3 kHz/ms)[15,35] and proposed that the direction selectivity existing in postsynaptic charges is simply inherited from the thalamus[8]. Why have previous studies not seen the enhancement of direction selectivity for slower FM sweeps? We believe that two key differences in circuit properties between awake and

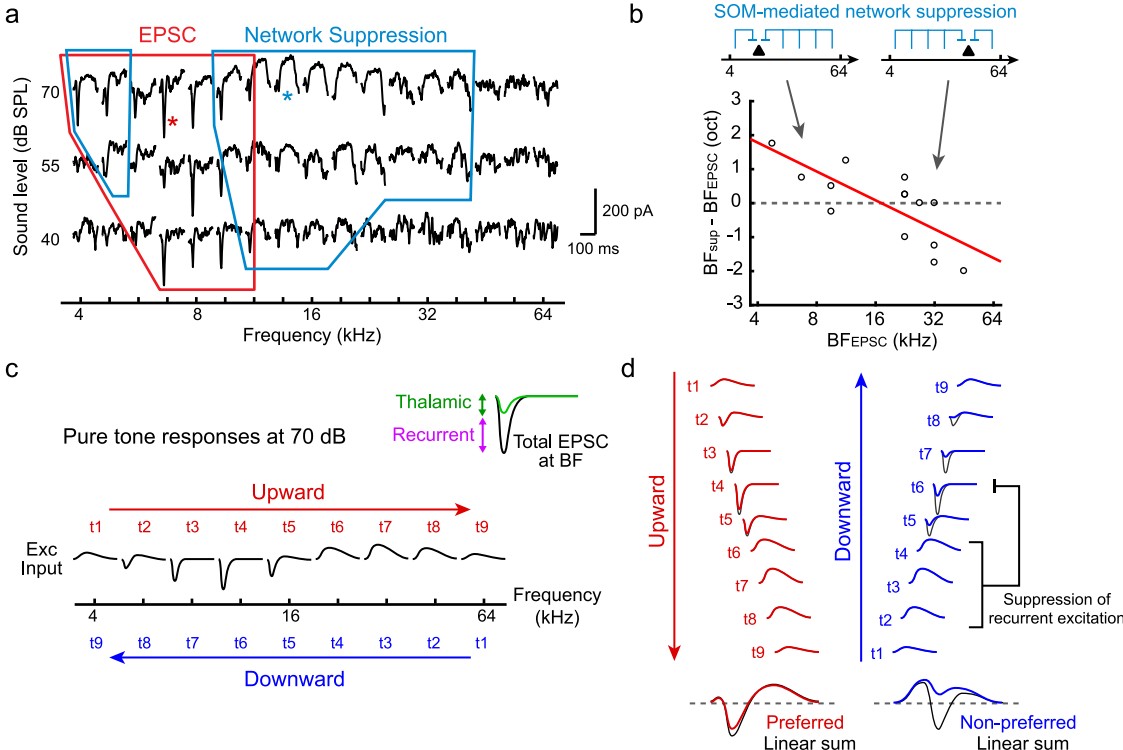

**Fig. 7 Simple schematic for the generation of direction selectivity by network suppression. a** TRF of EPSCs in a representative cell. Traces are an average of three trials. Red line outlines a region with fast EPSC, blue lines indicate regions with slow network suppression. **b** Neurons with low BF receive network suppression from high-frequency side, and those with high BF receive it from low-frequency side ($n = 14$ cells, $p = 0.0018$, two-sided $t$-test). **c** Schematic drawing of the TRF of EPSCs at one sound level. FM sweeps can be approximated as sequential presentations of pure tones. Inset shows the amplification of thalamic inputs by cortical recurrent circuits. **d** Schematic illustrating the generation of upward preference in a low-BF neuron. Top, pure tone responses from (**c**) temporally staggered to account for the frequency movement during FM sweeps. Bottom, compound EPSCs generated from upward and downward sequences of tones. In preferred direction, fast EPSC precedes network suppression, resulting in largely linear summation (red traces). In non-preferred direction, network suppression overlaps with fast EPSC and attenuates it, resulting in a sublinear summation (blue traces). Dark traces at the bottom show linear summation, which have equal total charges between directions.

anesthetized states could have contributed to these results. First, anesthesia is known to attenuate the activity of cortical SOM cells[49]. Since our optogenetic inactivation demonstrated a critical role for SOM cells in shaping direction selectivity, the absence of its activity under anesthesia may have simply diminished direction selectivity. Secondly, anesthesia dampens activity in many neural circuits and places neurons far below their spiking thresholds[50,51]. Theoretical studies have pointed out that a network can switch from ISN to non-ISN as the circuit activity level decreases[42,52]. Indeed, recent studies showed that the primary sensory cortex behaves as an ISN at rest in awake animals[22,25,27], whereas V1 under deep anesthesia operates as a non-ISN[53]. Thus, non-ISN operation of anesthetized brains in previous studies could have prevented the triggering of network suppression whose slow kinetics is crucial for generating direction selectivity. Our theoretical models support these two possibilities by demonstrating that both removals of SOM cell activity and attenuation of recurrent excitation (non-ISN) reduce direction selectivity. Furthermore, the range of FM rates where we observed enhanced direction selectivity in the awake state compared to the anesthetized state was consistent with our model (Fig. 4g and Supplementary Fig. 2). Overall, our data support the idea that A1 circuits in awake brains actively shape neuronal tuning to ethologically relevant FM sweeps, and that recurrent circuitry plays a critical role in this computation. Sensory integration via nonlinear recurrent dynamics may therefore be a general feature of cortical activity in awake brains.

**Cortical and subcortical sources of FM direction selectivity**. FM direction-selective firing has been previously observed in subcortical auditory structures, including cochlear nucleus[54], inferior colliculus[16–19], and thalamus[20]. In particular, direction selectivity in the inferior colliculus has been extensively studied in bats for its relevance in their echolocation and communication[55], and its generation is considered to involve temporal offsets between feedforward synaptic inputs[16,56]. Therefore, although we demonstrated SOM cell-mediated cortical enhancement of FM direction selectivity, it is likely that a part of direction selectivity in A1 is inherited from upstream peripheral structures[8]. This idea is consistent with the residual direction selectivity we observed after SOM cell inactivation (Fig. 3g), and both inheritance from upstream and recurrent circuit-mediated enhancement are compatible with asymmetry in FM sweeps-evoked postsynaptic charges (Fig. 2). In fact, assigning a moderate FM direction selectivity to the feedforward excitatory input in our network model supported the additive nature of direction selectivity from cortical and subcortical mechanisms (Supplementary Fig. 9d–g).

How neuronal computations in the sensory periphery contribute to cortical information processing has been debated across sensory modalities. Similar to our findings in the auditory system, generation of direction selectivity in the visual system has been reported both in the cortex[10,13] and peripheral retina[57]. Ablation of direction-selective retinal ganglion neurons in a previous study only partially reduced direction selectivity in the primary visual cortex, demonstrating additive contributions from both peripheral and cortical mechanisms[58]. Interestingly, this

retinal perturbation affected cortical responses mostly at high-speed visual motion (40º/s), suggesting that cortical mechanisms enhance visual motion processing predominantly at low- to mid-speed. Likewise, we found in our network model that the enhancement of FM direction selectivity within the A1 circuit is less prominent at high-speed FM sweeps (Fig. 4g). Thus, complementary extraction of rapid and slow time-varying stimuli in the periphery and the cortex—which likely relates to the loss of faithful encoding of rapidly varying stimuli at the cortex level[59,60]—may be a general feature shared across sensory modalities. Elucidating how the cortex inherits and transforms various sensory information from the periphery and expands encoding capacity will be a critical step in understanding cortical computations.

**Implications for FM sweeps processing in vocal communications.** The auditory cortex is necessary for behavioral discrimination of FM sweep directions[61,62]. Therefore, in order to understand perceptual behaviors triggered by FM sweeps in vocalizations, it is important to determine how FM direction information maps onto the cortical population activity. With some exceptions, most studies on FM direction selectivity have used long sweeps whose frequency covers nearly the entire range of mouse hearing (3–5 octaves). Due to this sound design, these studies assigned each cell a single DSI value, and direction selectivity has been topographically mapped onto A1 tonotopic axis such that low-A1 prefers upward sweeps and high-A1 prefers downward sweeps[15,34] (but the opposite in another study[12]). By contrast, we used ethologically more realistic FM sweeps with a restricted frequency range (1 octave) and found that direction selectivity is not a fixed property for each neuron but rather varies depending on the relative position of FM sweeps compared to the neuron's BF. This frequency-dependent direction selectivity has an obvious ethological advantage over fixed direction selectivity. In the latter system, neurons in high-A1 would mostly respond to downward sweeps and might fail to detect upward sweeps that are abundant in mouse ultrasonic vocalizations (Fig. 1). In contrast, in a system with frequency-dependent direction selectivity, FM sweeps can be encoded without losing information even in extremely high- or low-frequency domains, since each neuron takes part in the representations of both upward and downward FM sweeps. Together with our demonstration of direction-selectivity mechanisms for ethologically relevant slow FM sweeps, our findings show that A1 neurons in awake mice are able to encode rich information conveyed by FM sweeps in vocalizations. Since directions of slow FM sweeps provide critical cues to both phonemic identification[63] and lexical distinction[64] in humans, our results provide a step toward understanding neural circuits for vocal communications in more complex human brains.

## Methods

**Animals.** Mice were at least 6-week old at the time of experiments. Mice were acquired from Jackson Laboratories (VGAT-Cre, SOM-Cre, PV-Cre, ROSA-LSL-tdTomato, C57BL/6J, CBA/J, and BALB/cJ) or Charles River Laboratories (C57BL/6J, CBA/J, and BALB/cJ). Both female and male animals were used and housed in a reverse light cycle (12h-12h). All experiments were performed during their dark cycle. All procedures were in accordance with the Institutional Animal Care and Use Committee at the University of North Carolina at Chapel Hill and Osaka University as well as guidelines of the National Institute of Health.

**Mouse vocalization recordings.** Recordings were conducted from C57BL/6J, BALB/cJ, and CBA/J strains. Male courtship songs and pup isolation calls were recorded in a custom-built sound isolation chamber, using recording software Avisoft-RECORDER (Avisoft Bioacoustics, Glienicke, Germany) at a sampling rate of 250 kHz, microphone (CM16/CMPA; Avisoft Bioacoustics), and preamplifier (UltraSoundGate 116; Avisoft Bioacoustics). The microphone was placed 9 cm above the bottom of the cage. Courtship songs of male mice were triggered by placing a female mouse within the same cage. Pup calls, emitted when pups were isolated from their mothers, were recorded at the age of 8–10 days. Pain vocalizations were recorded in a sound-isolation chamber (Gretch-Ken Industries), using custom Matlab code for recording (Mathworks) at a sampling rate of 500 kHz,

microphone (4939-A-011; Brüel & Kjær), and conditioning amplifier (1708; Brüel & Kjær). The microphone was placed 5 cm above head-fixed mice. Pain vocalizations were triggered by mild electric shocks (0.5–0.8 mA, 0.5–1 s) at the tail.

**Intrinsic signal imaging.** Intrinsic signal images were acquired using a custom tandem lens macroscope and 12-bit CMOS camera (DS-1A-01M30, Dalsa). All mice were first implanted with a custom stainless-steel head-bar. Mice were anesthetized with isoflurane (0.8–2%) vaporized in oxygen (1 L/min) and kept on a feedback-controlled heating pad at 34–36 °C. The muscle overlying the right auditory cortex was removed, and a head-bar was secured on the skull using dental cement. The skull was kept transparent by saturation with phosphate-buffered saline. Mice were injected subcutaneously with chlorprothixene (1.5 mg/kg) prior to imaging. Images of surface vasculature were acquired using green LED illumination (530 nm) and intrinsic signals were recorded (16 Hz) using red illumination (625 nm). Each trial consisted of 1-s baseline followed by a 1-s stimulus (75 dB pure tone with a frequency of 3, 10, or 30 kHz, 10–20 trials for each frequency) and 30-s inter-trial interval. Images of reflectance were acquired at $717 \times 717$ pixels (covering $2.3 \times 2.3$ mm). Images during the response period (0.5–2 s from the sound onset) were averaged and divided by the average image during the baseline. Images were averaged across trials, Gaussian filtered, and thresholded for visualization. Individual auditory areas including A1, anterior auditory field (AAF), ventral auditory field (VAF), and secondary auditory cortex (A2) were identified based on their characteristic tonotopic organization.

**Two-photon calcium imaging.** Following the mapping of auditory cortical areas with intrinsic signal imaging in VGAT-Cre × Rosa-LSL-tdTomato mice, a craniotomy ($2 \times 3$ mm) was made over the auditory cortex, leaving the dura intact. Drilling was interrupted every 1–2 s and the skull was cooled with PBS to prevent damage from overheating. Virus (AAV9.syn.GCaMP6s.WPRE.SV40) was injected at 5–10 locations (250 μm deep from the pial surface, 30 nL/site at 10 nL/min). A glass window was placed over the craniotomy and secured with dental cement. Dexamethasone (2 mg/kg) was injected prior to the craniotomy. Enrofloxacin (10 mg/kg) and Meloxicam (5 mg/kg) were injected before mice were returned to their home cage. Two-photon calcium imaging was performed 2–3 weeks after chronic window implantation to ensure an appropriate level of GCaMP6s expression. Second intrinsic signal imaging was performed through chronic windows 1–3 days before calcium imaging to confirm intact auditory cortex maps. On the day of calcium imaging, mice were head-fixed in the awake state under the two-photon microscope within a custom-built sound-attenuating chamber. Responses to pure tones and FM sweeps were usually measured on two separate imaging sessions, with 59.3 ± 6.3% of cells imaged on both sessions. GCaMP6s and tdTomato were excited at 925 nm (InSight DS + , Newport), and images ($512 \times 512$ pixels covering $620 \times 620$ μm) were acquired with a commercial microscope (MOM scope, Sutter) running Scanimage software (Vidrio) using a ×16 objective (Nikon) at 30 Hz. Images were acquired from L2/3 (200–300 μm below the surface). Lateral motion was corrected by cross correlation-based image alignment[65]. Timings of sound delivery were aligned to the imaging frames by recording timing TTL signals in Wavesurfer software (Vidrio). Before each imaging session, tdTomato-expressing cells were identified by acquiring images in both green and red channels. The figure panel showing an example field of view (Fig. 1c) was generated by overlaying signals from two channels using Fiji software (https://imagej.net/Fiji).

**Surgery for in vivo electrophysiology.** After mapping auditory cortical areas with intrinsic signal imaging, the exposed skull was covered with silicone elastomer. One to five days later, mice were anesthetized with isoflurane and the skull was exposed by removing the silicone cover. A small (<0.3 mm diameter) craniotomy was made above A1 and a durotomy was made in most experiments. Special care was taken to reduce damage to the brain tissue during this surgery, since we observe abnormal activity from damaged tissue. We found it critical to interrupt drilling every 1–2 s and cool the skull with artificial cerebrospinal fluid (aCSF, in mM: 142 NaCl, 5 KCl, 10 glucose, 10 HEPES, 2-2.5 CaCl$_2$, 1-1.3 MgCl$_2$, pH 7.4) to prevent damage from overheating. Craniotomies were covered with aCSF and mice recovered from anesthesia for at least 1.5 h before electrophysiological recordings.

**In vivo whole-cell recording.** After the recovery from craniotomy and durotomy, mice were head-fixed in the awake state. During recording, mice sat quietly (with occasional bouts of whisking and grooming) in a loosely fitted plastic tube within a sound-attenuating enclosure (Gretch-Ken Industries). Whole-cell patch-clamp recordings were made with the blind technique. All recorded cells were located in L2/3, based on the z axis readout of an MP-285 micromanipulator (Sutter; 170–330 μm from the pial surface). Voltage-clamp recordings were made with patch pipettes (3.5-5 MOhm) filled with internal solution composed of (in mM) 130 cesium gluconate, 10 HEPES, 5 TEA-Cl, 12 Na-phosphocreatine, 0.2 EGTA, 3 Mg-ATP, and 0.2 Na-GTP (7.2 pH; 310 mOsm). EPSCs and IPSCs were recorded at −70 mV and +20 mV, near the reversal potentials for inhibition and excitation, respectively, set by our internal solution. Membrane potential values were not corrected for the 15 mV liquid junction potential and series resistance was continuously monitored for stability (average 26.4 ± 3.0 MOhm, n = 9 cells). Recordings were made with a MultiClamp

700B (Molecular Devices), digitized at 10 kHz (Digidata 1440A; Molecular Devices), and acquired using pClamp software (Molecular Devices).

In awake mice, we observed a continuous barrage of spontaneous EPSCs and IPSCs. Low basal activity with intermittent bursts was observed only in damaged preparations (e.g., bleeding brain tissue, tissue overheated during drilling, or too many electrode penetrations). Since detection of network suppression was critically dependent on the presence of intact spontaneous activity[22], recordings were not performed in mice with surgical damage. Experiments were terminated after 5–10 pipette penetrations when bursting activity appeared in recordings of field EPSPs.

**In vivo optogenetic inactivation during unit recording**. For inactivation of specific interneuron subtypes, AAV2/9.EF1a.DIO.eNpHR3.0.EYFP.WPRE.hGH was injected into the right auditory cortex of newborn *SOM-Cre* or *PV-Cre* mice (postnatal day 1–2). Pups were anesthetized by hypothermia and secured in a molded platform. The virus was injected at three locations along the rostral-caudal axis of the auditory cortex. At each site, injection was performed at three depths (1000, 800, and 600 μm deep from the skin surface, 23 nL/depth). Six weeks after injections, craniotomy and durotomy was performed as described above. After recovery from surgery, mice were head-fixed in the awake state. During recording, mice sat quietly (with occasional bouts of whisking and grooming) in a loosely fitted plastic tube within a sound-attenuating enclosure (Gretch-Ken Industries or custom-built). For unit recordings under anesthesia, mice were injected with 1.5 g/kg urethane and 1.5 mg/kg chlorprothixene prior to recordings, and body temperature was maintained at 36.0 °C using a feedback-controlled heating pad throughout the experiment. A 64-channel silicon probe (ASSY-77-H3, sharpened, Cambridge Neurotech) was inserted into A1. The probe was allowed to settle for at least 1 h before collecting data. Unit activity was amplified, digitized (RHD2164, Intan Technologies), and acquired at 20 kHz with OpenEphys system. A fiber-coupled LED (595 nm) was positioned 1–2 mm above the thinned skull and a small craniotomy. In interleaved trials, LED illumination was delivered that lasted from 500 ms before sound onset to 300 ms after the sound offset. Since we found that excessive inactivation of inhibitory neurons causes a paradoxical elimination of sound-triggered responses and often irreversibly changes the cortical activity to a bursty state, LED intensity was kept at the lowest effective intensity (1–3 mW/mm$^2$ at the brain surface). Before starting measurements of sound-evoked responses in each mouse, we monitored spontaneous activity without sound stimuli while applying brief LED illuminations, starting from low intensity and incrementally with higher intensities. We determined the lowest effective intensity that caused a visible increase in spontaneous activity and used that for the FM sweeps experiments. To ensure a similar amount of optogenetic effects between SOM and PV cell inactivation and to prevent irreversible changes in the cortical activity, experiments in which spontaneous firing rate is increased by 25–100% during LED illumination were accepted.

**Sound stimulus**. Auditory stimuli were calculated in Matlab (Mathworks) and delivered via a free-field electrostatic speaker (ES1; Tucker-Davis Technologies). Speakers were calibrated over a range of 2–64 kHz to give a flat response (±1 dB). Upward (4–64 kHz) and downward (64–4 kHz) logarithmic FM sweeps were presented at varying rates (2.5, 5,10, 20, 40, 80, 160 oct/s). In experiments for Fig. 6, one-octave FM sweeps (F$_{cent}$ at 5.7, 8, 11.3, 16, 22.7, 32, 45.3 kHz) were generated at the same FM rates. Sound stimuli had 3-ms linear rise-fall at onsets and offsets. Stimuli were delivered to the ear contralateral to the imaging or recording site. Auditory stimulus delivery was controlled by Bpod (Sanworks) running on Matlab.

**Analysis of FM rates in mouse vocalizations**. Syllables were extracted from recorded vocalizations using custom Matlab codes. Syllables were detected if the absolute amplitudes exceeded 10 × SD of the baseline noise. Onsets and offsets of syllables were determined using the criteria of 5 × SD, and syllables were dissected out with 15-ms margins on both ends. Contours of vocalization signals were separated from background noise by image processing of the spectrograms. For the analysis of male courtship songs and pup isolation calls, the spectrogram of each syllable was produced at 0.5-ms temporal resolution. Pixels with power above 25% (for male songs) or 20% (for pup calls) of the maximum value in each syllable were extracted as the signal. For the analysis of pain vocalizations, the spectrogram of each syllable was produced at 1–ms temporal resolution for B6 and BALB/c and 0.5-ms temporal resolution for CBA. For each time bin, broad band components were removed by subtracting the moving average (4000-Hz window) along the frequency domain. The spectrogram was further smoothened by using a median filter (3 × 3 pixels). The continuity of the contours was enhanced using the Frangi filter. Pixels with power above 5% of the maximum value in each syllable were extracted as the signal. After extraction of the contours of vocalization signals, a flood fill algorithm was used to isolate individual continuous components within a syllable (Supplementary Fig. 1). To further reduce noise, syllable components shorter than 3 ms (male songs and pup calls) or 7 ms (pain vocalizations) were rejected. Time and frequency values of individual components were obtained from the pixel positions. Each component was further divided into 5-ms segments, and FM rates were calculated for individual segments.

**Analysis of two-photon calcium imaging data**. Regions of interest (ROIs) corresponding to individual cell bodies were automatically detected by Suite2P

software and supplemented by manual drawing. However, we did not use the analysis pipeline in Suite2P after ROI detection, since we often observed over-subtraction of background signals. All ROIs were individually inspected and edited for appropriate shapes using a custom graphical user interface in Matlab. ROIs were aligned across days using affine transformation of the ROI positions, and ROIs from two days were judged to be the same cell if there is more than 60% overlap in the areas. Furthermore, a custom graphical user interface was used to visually inspect individual ROIs for the appropriate alignment. Pixels within each ROI were averaged to create a fluorescence time series F$_{cell\_measured}$(t). To correct for background contamination, ring-shaped background ROIs (starting at 2 pixels and ending at 8 pixels from the border of the ROI) were created around each cell ROI. From this background ROI, pixels that contained cell bodies or processes from surrounding cells were excluded. The remaining pixels were averaged to create a background fluorescence time series F$_{background}$(t). The fluorescence signal of a cell body was estimated as F(t) = F$_{cell\_measured}$(t) – 0.9 × F$_{background}$(t). To ensure robust neuropil subtraction, only cell ROIs that were at least 3% brighter than the background ROIs were included. Pure tone responses were measured during 1.2-s window after tone onsets. FM sweep responses were measured from sound onsets to 0.3 s after sound offsets, considering the slow kinetics of GCaMP6s. Cells were judged as significantly excited if they fulfilled two criteria: (1) dF/F had to exceed a fixed threshold value consecutively for at least 0.5 s in more than half of trials. (2) dF/F averaged across trials had to exceed a fixed threshold value consecutively for at least 0.5 s. The threshold for excitation (3.3 × standard deviations during the baseline period) was determined by receiver operator characteristic (ROC) analysis to yield a 90% true positive rate in tone responses. Two-photon imaging fields were aligned with the intrinsic signal imaging fields by comparing blood vessel patterns. Overall, 40% of GCaMP6s-expressing pyramidal cells (n = 1176 cells, 8 mice) increased their activity in response to at least one frequency. BF was calculated as the frequency with the strongest response independent of tone intensity.

Direction selectivity was determined using mean dF/F traces across at least five trials of presentations of each sound stimulus. DSI was calculated as (U−D)/(U + D), where U represents the response amplitudes triggered by upward FM sweeps and D represents those triggered by downward FM sweeps. For each ROI, DSI was calculated using only the FM rates that evoked significant excitatory responses in at least one direction. Response amplitudes were calculated as mean dF/F values during response measurement windows, and negative amplitudes were forced to zero in order to keep DSI range between −1 and 1. Response amplitudes were averaged across all included FM rates within upward or downward directions to calculate single DSI value for each ROI-FM range pair, except for the analyses where DSI for different FM rates was separately computed. Since our threshold yielding a 90% true-positive rate inevitably generates false-positive significant responses, we included in analyses only ROIs which are significantly responsive to at least three sounds. This substantially reduces the contamination from spontaneous significant responses that arise from random fluctuations while largely keeping real responses, since neurons with robust responses usually respond to multiple sounds. We also included results with lower threshold (including all responsive ROIs: Supplementary Fig. 4b) and with the higher threshold (including ROIs which are significantly responsive to at least five sounds: Supplementary Fig. 4d) in order to compare the DSI local heterogeneity between data selection criteria.

**Analysis of in vivo whole-cell recording data**. Data were analyzed using custom programs in Matlab. Response amplitudes were quantified from traces averaged across 3–9 trials of each sound stimulus. For pure tone (100 ms) responses, onset-locked EPSC (IPSC) was measured as negative- (positive-) going current 15–40 ms after tone onset, and network suppression was measured as positive- (negative-) going current 50–175 ms after tone onset. Cells were judged as responsive to specific combinations of frequency and intensity if they fulfilled two criteria: (1) the traces exceed a fixed threshold value (1 × standard deviation of baseline) consecutively for at least half the duration of detection window in more than half of trials, and (2) the averaged trace across trials exceeds a threshold value consecutively for at least half the duration of the detection window. BFs for onset-locked EPSC and network suppression were determined as the frequencies which evoked the strongest responses. For FM sweep (25–1600 ms) responses, to isolate fast synaptic currents from the overlapping slow network suppression, the trajectory of network suppression was first estimated by smoothing inactive portions of the trace using an iterative procedure. Fast EPSCs and IPSCs were detected when the trial-averaged trace deviates from the estimated network suppression trajectory by more than 3 × standard deviations during baseline. Once fast events were detected, onsets and offsets of currents were manually determined in a blinded manner with the assistance of a graphical user interface that automatically detects peaks and troughs of the traces. The excitatory (inhibitory) charge was calculated as the area below (above) the line which connects the onset and offset of the event. When multiple events were detected within a single trace, the charges were added together; however, the timings of the peak and the onset were calculated from the event with the biggest charge. EPSC-IPSC timing offsets were calculated only in cell-sound pairs where there were corresponding detected events in both EPSC and IPSC traces. DSI values were calculated in the same way as in two-photon calcium imaging experiments, except that the charges were included only up to 50 ms after the sweep offsets, considering the faster kinetics of electrophysiology (mean half-max duration of tone-evoked currents, EPSCs: 54.4 ± 8.8 ms; IPSC: 48.9 ± 6.5 ms).

**Analysis of single-unit recording data**. Single-units were isolated using Kilosort software (https://github.com/cortex-lab/KiloSort) and spike-sorting graphical user interface (Phy; https://github.com/cortex-lab/phy). Single-unit isolation was confirmed based on the inter-spike interval histogram (<3% of the spikes in the 2-ms bin, after correction for the overall spike frequency) and the consistency of the spike waveform. Positions of the cortical surface, layers, and white matter were identified by current source density analysis and the distribution of multi-unit spikes. Fast-spiking units were identified based on their small trough-peak interval (<0.4 ms). To quantify FM sweeps-evoked responses, PSTH was generated at 50-ms bin. Sound-evoked responses were quantified after subtraction of the baseline firing rate 0–200 ms before sound onsets. Thus, sound-evoked responses do not include the increase in the spontaneous firing rate caused by LED. Units were judged as significantly excited if they fulfilled two criteria: (1) PSTH had to exceed a fixed threshold value at the same time bin in more than one third of trials. (2) Trial-averaged PSTH had to exceed a fixed threshold value. The threshold for excitation ($3.7 \times$ standard deviations during the baseline period before LED) was determined by ROC analysis to yield a 90% true positive rate in tone-evoked responses. DSI values were calculated in the same way as in whole-cell recording experiments.

**Construction of network models**. To model A1, we organized neurons along a one-dimensional spatial continuum over the tonotopy. Let $2^x$, with $x \in [2,6]$ denote the BF (in kHz) of a neuron. We consider a phenomenological spatial firing rate model,

$$\tau_\alpha^m \frac{dA_\alpha}{dt} = -A_\alpha + \sum_\beta \int K_{\alpha\beta}(x, y) F\left(r_\beta(y)\right) dy + b_\alpha(x, t) + \mu_\alpha \qquad (1)$$

where $A_\alpha$ denotes the average activity of inputs to population $\alpha = E, P, S$, the integral is over the whole tonotopic axis, and the sum is over all populations. These inputs are fed through a temporal filter of the form

$$\tau_\alpha^r \frac{dr_\alpha}{dt} = A_\alpha - r_\alpha \qquad (2)$$

which is used to account for additional population-specific synaptic and membrane properties in the conversion from the activity of inputs to firing rate. Motivated by experimental results showing the slow recruitment of SOM neurons[66,67], for simplicity, we only consider $\tau_s^r \neq 0$ while taking the other two filters to be instantaneous. However, we note that similar results hold true for $\tau_E^r$, $\tau_p^r \neq 0$, though this added oscillatory behavior in the return to steady state and added numerical complexity. Lastly, the firing rate function $F(x)$ is taken to be

$$F(x) = [x]_+ = \begin{cases} x & \text{if } x > 0 \\ 0 & \text{otherwise} \end{cases} \qquad (3)$$

to prevent negative firing rates. The connection strength from population $\beta \to \alpha$ at position $y$ and $x$, respectively, is given by

$$K_{\alpha\beta}(x, y) = W_{\alpha\beta} e^{-(x-y)^2/\lambda_{\alpha\beta}^2}/G(x), \qquad (4)$$

where the normalization term is chosen such that there exists a spatially uniform steady state, namely

$$G(x) = \int e^{-(x-y)^2/\lambda_{\alpha\beta}^2} dy. \qquad (5)$$

Specifically, this choice ensures that at each point in the tonotopy, the total effective interaction to population $\alpha$ from population $\beta$ is $W_{\alpha\beta}$. We take the effective connectivity between two neurons to decay as a Gaussian in the difference between the neurons' BFs in octaves. In the absence of auditory inputs each of the populations in our model has a spatially uniform steady-state firing rate. The baseline background activity to each population is $\mu_\alpha$. The stimulus input to population $\alpha$ is taken to be

$$b_\alpha(x, t) = \text{Amp}_\alpha e^{\left(x - \left(st + \log_2 f_0\right)\right)^2/\sigma_\alpha^2} \qquad (6)$$

where $s$ is the sweep speed ($s > 0$ for upward sweep), and $f_0$ is the starting frequency of the sweep. Lastly, the DSI for the mathematical model is

$$\text{DSI} = \frac{\int_0^\infty \Delta r_E^{up} dt - \int_0^\infty \Delta r_E^{down} dt}{\int_0^\infty \Delta r_E^{up} dt + \int_0^\infty \Delta r_E^{down} dt} \qquad (7)$$

where $\Delta r_E$ represents the change in firing rate from baseline

$$\Delta r_E = F(r_E) - r_E^0. \qquad (8)$$

The default parameter values for the model can be found in Supplementary Tables 1–3. The photoinactivation simulations were done by weakening the effective connectivity of each inactivated population by 20% and introducing an inhibitory current of strength $-1.5$ into their respective $dA_\alpha/dt$ equations. The system was allowed to reach its new steady state before the FM sweep stimulus. The shift from ISN to non-ISN was achieved by decreasing the effective interaction strength matrix and feedforward connections by multiplication with $\beta = 0.5$ or $0.2$ and decreasing the baseline firing rate of pyramidal cells to be $r_E^0 = 0.1$ Hz. The tonotopic axis was discretized into 401 points with $x \in [2, 6]$. The lowest and highest edges of the tonotopic axis were excluded from the data due to the oscillation generated at the boundaries. All simulations were completed using ode45, a numerical solver for differential equations implemented by Matlab, with maximum relative tolerance and absolute error levels set to $1 \times 10^{-5}$ and $1 \times 10^{-6}$, respectively.

For the figure examining the impact of direction selectivity in the feedforward input, the amplitude of the feedforward input was adjusted in the following manner

$$\text{Amp}_\alpha(t) = \begin{cases} 0, & \text{for } t < t_{start} \text{ and } t > t_{stop} \\ \text{Amp}_{max} + \frac{\text{Amp}_{min} - \text{Amp}_{max}}{t_{stop} - t_{start}} \left(t - t_{start}\right) \end{cases} \qquad (9)$$

where $\text{Amp}_{min}$ is 0.8 $\text{Amp}_{max}$, and $t_{start}$ and $t_{stop}$ are the start and stop times of stimulus, respectively.

**Statistical analysis**. All data are presented as mean ± SEM. Statistically significant differences between conditions were determined using standard parametric or nonparametric tests in Matlab. In cases where parametric statistics are reported, data distribution was assumed to be normal, but this was not formally tested. $R$ values were calculated as Pearson's correlation coefficients, and their statistical significance was computed by transforming the correlation to create a t-statistics in Matlab corrcoef function. Randomization is not relevant for this study because there were no animal treatment groups. All $n$ values refer to the number of cells except when explicitly stated that the $n$ is referring to the number of mice or number of cell-sound pairs. Experiments were not performed blind. Sample sizes were not predetermined by statistical methods but were based on those commonly used in the field. The number of data points, $p$ values, and statistics used in individual figures are listed in Supplementary Data 1.

**Reporting summary**. Further information on research design is available in the Nature Research Reporting Summary linked to this article.

## Data availability
Source data for all figures are provided as a supplementary data file with this paper. Other data that support the findings of this study will be made available from the corresponding author upon reasonable request. Source data are provided with this paper.

## Code availability
The code for all simulations can be found on GitHub (https://github.com/gregoryhandy). Other custom Matlab codes used in this study will be made available from the corresponding author upon reasonable request.

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

## Acknowledgements

We thank Douglas Kim and the GENIE Project at Janelia Research Campus for making GCaMP available, Arikuni Uchimura for assistance with vocal recording, Paul Manis, Thanos Tzounopoulos, Toshihide Hige, Andrea Giovannucci, and Jose Rodriguez-Romaguera for comments on the manuscript. This work was supported by NIDCD (R01DC017516), Pew Biomedical Scholarship, Whitehall Foundation, Klingenstein-Simons Fellowship (H.K.K.), Foundation of Hope (H.K.K. and H.T.), NINDS (F31-NS111849, T32-NS007431; A.M.K.), and NIH grants 1U19NS107613-01 and R01 EB026953, the Vannevar Bush Faculty Fellowship #N00014-18-1-2002, and a grant from the Simons foundation collaboration on the global brain (B.D.).

## Author contributions

D.A.A., A.M.K., and H.K.K. designed the project, conducted in vivo experiments, and analyzed data. G.H. and B.D. designed and programmed the network model. H.T. performed the vocal recordings and analysis. D.A.A., G.H., B.D., and H.K.K. wrote the manuscript, with inputs from A.M.K. and H.T.

## Competing interests

The authors declare no competing interests.
