## [Peer Review File · Nature Communications]

Reviewers' Comments:

Reviewer #1:

Remarks to the Author:

Report on "Recurrent network dynamics shape direction selectivity in primary auditory cortex" by Aponte et al. Nature Communication, NCOMMS-20-10199-T, 2020

Neurons in the primary auditory cortex (A1) exhibit direction selectivity, i.e., they fire selectively to upward or downward frequency modulations (FM). This direction selectivity (DS) is classically attributed to temporal offsets between feedforward excitatory and inhibitory inputs. In this study the authors investigate an alternative hypothesis, namely that the recurrent cortical dynamics is critical for direction selectivity in A1. Using a combination of two-photon calcium imaging, whole-cell recordings in awake mice and computational modeling they demonstrate that the cortical recurrent dynamics play a pivotal role in the emergence of DS in contrast with the current classical view. In particular they show that upon optogenetic inactivation of SOM interneurons reduces DS, revealing its cortical origin.

The results of this paper are important for our understanding of the network mechanisms underlying the processing of complex auditory stimuli. They have an even broader relevance since direction selectivity is also observed for other sensory modalities e.g. in visual and somatosensory cortices. Moreover, this paper overall is well written. In particular, although I am a theoretician, I found the sections dedicated to the experimental results easy to follow. Therefore, it fits well for the general audience of Nature Communication.

My main points will be on the modelling part.

1- The model: Eq., line 605, describes a spatial firing rate model linearized about its baseline firing rate. Later on, the authors show that nonlinearities are required to explain the experimental data. To this end they introduce an ad-hoc threshold (line 613-616). I found this a bit confusing. Why not considering from the beginning a non-linear model with a threshold-linear non-linearity ?

2-Interpretation of the rate model: Rate models can be interpreted in two ways: 1) as phenomenological models in which the dynamics describe the time evolution. In that case the time constants which characterise the dynamics are phenomenological parameters whose physiological meaning are not clear; 2) as a spiking network model in the limit of infinitely slow synaptic dynamics. In practice such models provide a good description of spiking network provided that the single neuron dynamics is sufficiently fast compared to the synaptic dynamics.

Here, the model is described as in 1) whereas the interpretation 2) seems to me to fit better to interpret the experimental results. I suggest to the authors to adopt the latter interpretation and revise the paper in accordance (in particular the parameters $[\tau_E, \tau_P, \tau_S]$ should be interpreted as synaptic time constants and their values chosen in agreement with synaptic physiology); see also 4. .

3-The slow time scale in Eq. 609 for SOM neurons is essential for the model to explain the data. I may have missed it but I could not find a physiological justification for the value - 100msec - that is assumed for this parameter.

4-On the same token, I could not find a justification for the values chosen for the rate time constant of $[\tau_E, \tau_P, \tau_S] = [10, 20, 20]$ (ms)

5-Why considering an extended 1-D model with many columns ? Is it really necessary for the argument? It seems to me that a model involving only two columns would be sufficient. If the authors agree with that, I suggest to focus on such a model in the paper and to move the results of the full

model in Supplementary data.

6-The ISN regime: The authors write (line 247-252) ‘‘ Lastly, we investigate how a shift from an ISN to non-ISN regime, a change potentially resembling a shift from awake to anesthetized states (see Discussion), affects these results [...]. We find that weakening of recurrent connectivity is enough to push the system to operate in a linear regime and eliminate both network suppression and direction selectivity (Fig.5d). Then in line 286-287 ‘‘In this study, we further demonstrate that network suppression in ISNs plays a critical role in generating direction selectivity to ethologically meaningful slow FM sweeps.’’ I do not find the argument that the network must an ISN compelling since the authors present their results for one set of parameters. As a matter of fact, my intuition of the mechanism does not require ISN but only slow time constants and a non-linearity. The authors should elaborate more on their argument for ISN.

11111
SEP:SEP

Reviewer #2:

Remarks to the Author:

Aponte et al. address the relevant question how the selective tuning of neurons in the auditory cortex to the direction of frequency modulated sweeps is computed. The currently prevailing opinion is that direction tuning in the cortex is mostly reflecting tuning from feed-forward inputs from the thalamus. They show that this class of stimuli are ethologically relevant for different strains of mice and using awake in vivo calcium imaging and electrophysiology techniques, they show that a large population of cortical neurons are responsive to frequency modulated stimuli in a direction selective manner. Interestingly, they demonstrate that the underlying conductances are essentially larger for the preferred direction as for the non-preferred directions, but do not show major differences in their latencies. Furthermore, the authors investigate whether direction selectivity is computed in recurrent cortical circuits by applying optogenetic manipulations of different populations of inhibitory neurons. Moreover, they develop a network model consistent with an essential computation of direction selectivity within cortical circuits capturing some of the observed experimental findings. Although, the question is obviously interesting and relevant, technically demanding experiments were performed and their main conclusion that this particular tuning property is calculated at the cortical level is probably right, I do, unfortunately, not see that their data and modeling support this conclusion convincingly. I am therefore not in favor of publication of the manuscript in Nature communications.

The main concern is as follows:

The authors claim that they ‘‘reveal a major contribution of recurrent network dynamics in shaping cortical tuning’’ (namely direction selectivity). In Figure 1h, they summarize their recordings by plotting a direction selectivity index (DSI, bound from -1 to 1) over the BF of the recorded cells. These indices are hugely variable, independent of the BF. The data shown in 1d suggests that this heterogeneity reflects biology and is not just ‘noise’. When regressing the data, an anticorrelation between DSI and BF becomes apparent. Indeed, the data looks as if there is a bias on the neurons’ direction selectivity depending on where they are positioned on the tonotopic map, perhaps hinting at some ‘edge effect’.

Importantly, however, this effect is only very small. If one would subtract this global bias from the data in Figure 1h (or would subtract the prediction from the model as shown in Figure 4f), the plot would essentially still look the same and only a minor fraction of the variability that is observed at a given BF-bin is explained in the end. However, the rest of this manuscript essentially deals with this small global effect. It does not provide insights how the pronounced and highly heterogeneous tuning properties of a set of neurons from the same position along the tonotopic axis in the end emerges. Also, the proposed network model does not capture this heterogeneity and solely reproduces the

gradient in overall tuning (Figure 4f).

When focusing on global effects in direction tuning biases, the experimental design, however, is complicated, as the hearing sensitivity in mice is most sensitive around 8-16kHz and decreases towards the edges of the hearing range. When delivering sweeps that were calibrated to maintain a constant intensity, they will nevertheless, due to the different sensitivities of the auditory system, provide an input to the cortex that changes frequency and intensity at the same time. However, it has been previously demonstrated that sounds with the same spectral content, but increasing or decreasing intensity envelope, also induce highly non-linear and asymmetric responses in the mouse auditory cortex (Deneux et al. 2016, Nat. Comm.). This confound, and how it could relate to the observations, is not mentioned at all in the manuscript.

Additional points (major):

1) In Figure 3g, the authors present the change in DSI of recorded units with and without manipulation of SOM+ and PV+ cells, respectively. They categorize the recorded units into sweep responsive, and direction selective units. While it seems that every direction selective unit should by construction also be sweep responsive, it is difficult to interpret the data as presented. Specifically, in control conditions without light, why do some 'direction selective' units (red) have DSI indices of ~ 0 ? Related, why do sweep responsive, but 'non direction selective', units (gray) have DSI indices of ~ 1 ?

2) The authors cite a number of studies well known in the field suggesting a sub-cortical origin for direction tuning. They claim that the contradicting observations in the current study were due to different experimental conditions, i.e. anaesthetized vs awake mice. However, to support this important claim, and to resolve this apparent contradiction, it would be crucial to compare these conditions side-by-side.

3) In optogenetic experiments a prime parameter is the intensity of the light that is being applied, as the observed effects may critically depend on it. Here, the authors normalize the inhibition by PV+ and SOM+ cells by normalizing to spontaneous output firing of excitatory neurons (Figure 3h). It seems unclear why 1.5-fold increased spontaneous firing is used for normalization. The effect on the spontaneous activity levels may depend on qualitative differences between PV+ and SOM+ cells as they may have a qualitatively different influence on direction selectivity (as the authors report). However, this is an arbitrary approach. To illustrate this, just consider the inverse, such that the light intensities would have been first adjusted such that PV+ and SOM+ cells would have a similar effect on direction tuning, and then the differential effects on spontaneous firing rates would have been described. Demonstrating the robustness of the effects across a wide range of light intensities would help convince the reader that the effects are indeed dependent on the targeted cell-type and not on the specific light intensity settings.

Additional points (minor):

1) It is unclear whether the data presented in Figure 1h stems from all tested rates or not. Considering the importance of rate in many other quantifications, this is confusing.

2) The title of Figure 2 highlights the asymmetry of excitatory inputs between FM directions, which the presented data clearly supports. But why do the authors focus on excitation, when inhibitory inputs seem to be equally asymmetrically tuned (Figure 2d left)?

3) In the model, there seems to be a strong dependence of absolute rates of FM sweeps on DSI (Figure 4g). This is not the case for the experimental data (Figure 1f or Figure 2e).

4) There are different meanings of the acronym TRF, it should be defined in the manuscript.

5) Throughout the text statistical p-values are given, however, no further information on what exactly was tested, e.g. what test was being applied.

6) Figure 5a illustrates the effects on direction tuning when simulating the optogenetic manipulations in the network model. Here, the middle panel showing the firing rate in response to upward or downward sweeps still reveals a substantial asymmetry (alike Fig. 4dv), although the claim would be that this difference would vanish upon the block of SOM+ cells.

Reviewer #3:

Remarks to the Author:

Aponte and co-authors combine awake calcium imaging, in vivo whole cell voltage clamp recordings, computational modeling and single unit recordings with optogenetic silencing to challenge a prevailing model of how auditory cortex circuits shape the encoding of frequency modulated (FM) sweep direction. They provide compelling evidence that somatostatin-positive GABA neurons suppress firing at non-preferred directions through network suppression – a neurophysiological mechanism shown previously by their lab to sculpt sideband inhibition. Overall, this is a really beautiful study. The data address the narrow question of how FM direction is encoded in the mouse auditory cortex but also extends to bigger issues, including how cardinal features of cortical organization including inhibition stabilized networks and extensive recurrent connectivity can shape feature selectivity in ways that would not be seen in subcortical areas. I see only one substantial issue with their manuscript that is easily fixed by text changes and possibly a few additional changes to their model.

General points:

1. They over-interpret their data when they write, “FM direction selectivity emerges in cortical circuits” (ln 201), “Our SOM cell photoinactivation experiments demonstrated the generation of direction selectivity to slow FM sweeps within the cortex (ln 335-6), “revealing its cortical origin” (ln 33). Subcortical neurons clearly have FM direction selectivity has evidenced by dozens of papers documenting the phenomenon in the inferior colliculus and medial geniculate body (and in the brainstem as well). FM direction selectivity definitely DOES NOT originate in the cortex and I think choice of wording will really hit a nerve with readers. What they data do show – very elegantly and convincingly – is that SOM cells greatly enhance / strongly shape etc. FM selectivity at the level of the cortex. This can be aI would like to see an expanded section of the Discussion that addresses the point that FM direction selectivity is already constructed subcortically. As it stands, they cite reference 8, which did not even record from a subcortical structure. For example, it would be important to consider their findings in relation to George Pollack’s work on FM direction coding in the inferior colliculus (yes, it is a bat but it’s not an echolocating bat, so it’s basically the same as a mouse).

Their over-interpretation comes from a combination of not citing the subcortical literature but also over-interpreting the lack of fast EPSC/IPSC for non-preferred directions. 1) Their assertion that thalamic inputs have no direction selectivity is based on recordings from L2/3 pyramidal neurons and as such is not the basis for ruling out what the cortex can receive from the thalamus, 2) the fast EPSC/IPSC does not definitely isolate a thalamocortical contribution, 3) even their favorite mechanisms, recurrent activity, can depend critically on ongoing input from the thalamus (see Reinhold and Scanziani) and they have not ruled out that thalamic inputs that either target or disproportionately drive SOM neurons is not critical for findings they report here. In future work, they could use 2-p imaging of MGB thalamocortical terminals or silence cortical activity pharmacologically or with PV-ChR2 while recording EPSCs across middle layers to isolate what the MGB is transmitting to the cortex.

Apart from backing off of their wording and adding citations to the Discussion, I am concerned that

their model parameters are not set realistically because they assume the DSI for thalamic inputs is zero. Could a supplemental figure address how slight increases in thalamic DSI would affect their findings?

My only motivation here is to help the authors showcase their work in a way that will resonate with the community of researchers that work on the auditory system. Clearly, I am a fan of this paper and admire the work that this lab is publishing. If they back off the language that direction coding originates in the cortex, do a better job of citing the literature that demonstrates direction coding subcortically, and include a portion of the Discussion that considers the cortical circuit mechanism to what is constructed subcortically and what is passed on through thalamocortical synapses (the two are not the same), I think this beautiful paper will get the higher citation counts that it deserves.

2. For the optogenetics experiments, they should clarify what type of units they are recording from. Presumably, they could sort their sample into regular spiking and fast spiking based on the spike waveform. Further, are they able to identify SOM neurons based on spiking changes arising directly from activating the opsin?

Either way, to make the single unit recordings more comparable to the model they could focus only on the RS putative pyramidal units. If they have enough FS/PV units, they could provide a supplemental figure to show how SOM inactivation affects PV DSI and how this is predicted by the model. Obviously, if they have enough optogenetically identified SOM neurons, it would be hugely important and interesting to show that they have the broad FM selectivity assumed by the model (e.g., in Fig. 4d, iv).

It would be helpful if some of the points above could be addressed in a supplemental figure alongside more examples of single units recorded in the two optogenetic silencing conditions. The two examples in Fig. 3 have very different firing rates and it is unclear why the evoked response occurs at the same time for Pref and Non-Pref in 3C unless this cell is tuned to ~16kHz or has much broader frequency tuning. For this reason, it would be helpful if they could indicate with an arrow in all of their FM-evoked activity traces when the sweep crosses through the BF/TRF of the neuron or – better yet – include the tonal receptive field for each of the units they show in this figure and elsewhere.

Specific points:

3. Could they clarify whether all recordings were made in the left hemisphere, as suggested in Fig. 1? Hemispheric differences in FM sweep coding have been reported previously and based on this literature the right hemisphere would be expected to show stronger direction coding.
4. It would be helpful if the authors provided a cartoon schematic for the prevailing model (FM direction created by fast IPSC timing) and the SOM/ISN model proposed here. It would be nice if this were included at the end of Fig. 1 or the beginning of Fig. 2 to put the data into context.
5. Could the ISN model also capture the attenuation of fast IPSCs during the non-preferred direction of FM sweeps?
6. What are the temporal offsets of the EPSCs and IPSCs from the ISN model? Does it also show a comparable timing of EPSCs and IPSCs?
7. In the framework of network suppression by the non-preferred direction of FM sweep, the schematic (Supplementary Figure 5C) illustrates how the network suppression can recapitulate the direction of selectivity based on the central frequency of FM sweeps and BF of a neuron. Could the ISN model in Figure 4 and 5, predict the phenomenon in Figure 6?
8. Figure 4d, subpanel v shows the firing rate of a PyrN from the model for preferred and non-preferred directions. Figure 5a shows the firing rate of a PyrN with the same BF, only this time with a partial block of SOM inputs. The issue is that both of these plots look nearly identical and it is unclear why/how that would be.
9. On Ln 259 they mention that shorter FM sweeps are prevalent in mouse vocalizations. It seems they have the acoustic recordings to substantiate this claim but the evidence is not shown. Can it be provided? Further, I suggest that they use the term "restricted" FM instead of "shorter" bcs the latter

is confusing about whether they refer to time or frequency.

Minor points:

10. Line 131: when first mention TRF, please include the full name of the abbreviation.

11. Lines 139-140 – I disagree with their statement, “consistent with the idea that inhibition is recruited through recurrent excitation within A1”. Although I think the SOM inhibition data they present later supports this view, at this point in the manuscript the findings they observe could be explained equally well by DSI that is entirely inherited from upstream neurons that have direction selectivity. In either case, you would have no EPSC/IPSC for the non-preferred direction. In fact, they could use this ambiguity as the motivation to perform the optogenetics experiment because it would make for a nice transition.

12. Line 641: what is ode45?

13. Line 653: “The code for all simulations can be found on GitHub (<https://github.com/gregoryhandy>)”. The code is not yet on the GitHub site.

14. As a technical curiosity, does 615nm excite tdTomato? How does increased endogenous fluorescence affect the ISI quality?

15. Looking at the asterisks in Fig. 2b, it’s not clear how they operationally define the contribution of network suppression. How do they distinguish between sound-evoked synaptic currents that end versus sound-evoked synaptic currents that are actively suppressed? In both cases, the currents would be reduced.

16. The writing style for the last sentence on Pg. 6 (Ln 126-129) could be improved. It’s a run-on in its current form.

17. Line 167, “We used weak LED intensities to modulate transmission 167 without inducing aberrant discharges” and Ln 487 “to prevent aberrant sound-evoked cortical activity”. They only take units where the firing rate was increased by 25-100% during illumination (489-90), which by definition is aberrant (i.e., not normal). They are changing the spike rates by inactivating PV and SOM neurons so they should not try to cover this up. Moreover it isn’t clear why they report the power as 10-35 mW. Is this at the fiber tip? Why not express as mW/mm² as measured on the brain? Lastly, how was the “lowest effective intensity” determined and can this be appended to the Methods?

18. Line 257, pure tone FM sweeps are not ethologically meaningful. The rates of FM they use are ethologically relevant. I suggest the wording be changed.

19. Line 410, the preparation of the skull for ISI is not clear. Is the skull intact? Thinned? Made transparent by wetting it with saline?

20. Line 432, How are the images and cells aligned for the two imaging sessions to confirm that pure tone and FM measurements are made from the same cells?

21. Line 474, why do they inject the eNpHR3 virus in newborn mice instead of young adults like the GCaMP?

22. Line 485, what do they mean thinned skull? They made a craniotomy so isn’t the light from the LED directed at the exposed brain?

Response to Reviewers

Reviewer #1:

We are delighted this reviewer found that *“The results of this paper are important for our understanding of the network mechanisms underlying the processing of complex auditory stimuli. They have an even broader relevance since direction selectivity is also observed for other sensory modalities e.g. in visual and somatosensory cortices. Moreover, this paper overall is well written. In particular, although I am a theoretician, I found the sections dedicated to the experimental results easy to follow. Therefore, it fits well for the general audience of Nature Communication.”* Below are the responses to this reviewer’s comments:

“My main points will be on the modelling part.

1- The model: Eq., line 605, describes a spatial firing rate model linearized about its baseline firing rate. Later on, the authors show that nonlinearities are required to explain the experimental data. To this end they introduce an ad-hoc threshold (line 613-616). I found this a bit confusing. Why not considering from the beginning a non-linear model with a threshold-linear non-linearity ?

2- Interpretation of the rate model: Rate models can be interpreted in two ways: 1) as phenomenological models in which the dynamics describe the time evolution. In that case the time constants which characterise the dynamics are phenomenological parameters whose physiological meaning are not clear; 2) as a spiking network model in the limit of infinitely slow synaptic dynamics. In practice such models provide a good description of spiking network provided that the single neuron dynamics is sufficiently fast compared to the synaptic dynamics.

Here, the model is described as in 1) whereas the interpretation 2) seems to me to fit better to interpret the experimental results. I suggest to the authors to adopt the latter interpretation and revise the paper in accordance (in particular the parameters $[\tau_E, \tau_P, \tau_S]$ should be interpreted as synaptic time constants and their values chosen in agreement with synaptic physiology); see also 4.”

We thank the reviewer for highlighting their confusion with regards to how we introduced the model in the main text, as well as their interpretation of the equations. Taking the reviewer’s suggestion, we now introduce the model by considering the non-linear model and adjusted Fig. 4c accordingly. This is now explicitly stated in our revised manuscript as

“A threshold-linear firing rate function was used to prevent the firing rates from going below zero (Fig. 4c).”

The paragraph outlining the importance of the non-linear threshold was moved to the paragraph after the discussion of the modeling results found in Fig. 4:

“Is the threshold nonlinearity we introduced in the model, as opposed to a completely linear model, required to capture direction selectivity? This influence of the threshold can

easily be removed by increasing the baseline firing rate of excitatory neurons, leading to a completely linear network model. Interestingly, if this is the case, we can prove that DSI = 0 for all choices of model parameters (see Supplementary Text). This important insight shows that, in addition to modeling the proper circuit structure, we must also consider a nonlinear neuronal transfer function in order to explore the mechanistic underpinnings of direction selectivity.”

Furthermore, to address both points 1 and 2 of this reviewer, we revised our model, explicitly including the non-linearity in the equation, as you see in the Methods section of our revised manuscript:

“We consider a phenomenological spatial firing rate model,

$$\tau_{\alpha}^m \frac{dA_{\alpha}}{dt} = -A_{\alpha} + \sum_{\beta} \int K_{\alpha\beta}(x, y) F(r_{\beta}(y)) dy + b_{\alpha}(x, t) + \mu_{\alpha}$$

where A_{α} denotes the average activity of inputs to population $\alpha = E, P, S$, the integral is over the whole tonotopic axis, and the sum is over all populations.”

In the original draft, it was unclear whether the non-linear firing rate function $F(x)$ appeared inside or outside of the summation, which, we believe, led to the confusion over how one should interpret this equation, as the reviewer pointed out. As derived in Ermentrout and Terman (Ermentrout GB, Terman D. Firing Rate Models. In: *Mathematical Foundations of Neuroscience*. Springer, 2010, p. 331–367), if the function $F(x)$ had appeared on the outside, the reviewer’s suggestion of interpreting the equations as synaptic drive in the limit of infinitely slow synaptic dynamics would be correct. With it appearing on the inside, it is more appropriate to stick with the phenomenological interpretation. We have also chosen to relabel r_{α} , which originally appeared in this equation, as A_{α} to further illustrate this chosen interpretation. Further, we relabeled the temporal filter (i.e., z_{α} in the original draft) to be r_{α} .

In this form, the model appears to have several additional variables not originally specified, namely μ_{α} , the baseline background activity, and r_P^0 and r_S^0 , the steady state firing rate of PV and SOM neurons. These parameters are in fact superfluous with respect to the results of this paper and DSI, and this is explicitly shown in a new “Examining the change in baseline with the non-linear model” section in Supplemental Text:

“Since DSI requires only the change in baseline firing rate, the parameters μ_{α} superfluous to this study. To see this, let r_{α}^0 denote the steady state firing rates of the system when $b_{\alpha}(x, t) = 0$ and denote the change from baseline as ΔA_{α} . Substituting $\Delta A_{\alpha} + r_{\alpha}^0$ into our ODE for ΔA_{α} yields

$$\tau_\alpha^m \frac{d(\Delta A_\alpha + r_\alpha^0)}{dt} = -(\Delta A_\alpha + r_\alpha^0) + \sum_\beta \int K_{\alpha\beta}(x, y) F(\Delta r_\beta(y) + r_\alpha^0) dy + b_\alpha(x, t) + \mu_\alpha.$$

This equation can then be simplified and rewritten as

$$\begin{aligned} \tau_\alpha^m \frac{d\Delta A_\alpha}{dt} = & -\Delta A_\alpha + \sum_\beta \int K_{\alpha\beta}(x, y) \hat{F}_\beta(\Delta r_\beta(y)) dy + b_\alpha(x, t) \\ & + \left[\sum_\beta W_{\alpha\beta} r_\beta^0 + \mu_\alpha - r_\alpha^0 \right], \end{aligned}$$

where

$$\hat{F}_\beta(x) = \begin{cases} x & \text{if } x > -r_\beta^0 \\ -r_\beta^0 & \text{otherwise} \end{cases}.$$

Since r_α^0 is the steady state solution, the terms in the bracket sum up to zero, eliminating the parameters μ_α from appearing. Furthermore, the baseline firing rates only appear in the adjusted thresholding function $\hat{F}_\beta(x)$. We show that for DSI to be non-zero, this non-linear threshold must come into effect for at least the E populations. As a result, for simplicity we assume that the PV and SOM equations operate solely in the linear regime, resulting in r_P^0 and r_S^0 also being superfluous parameters that need not be specified.”

“3-The slow time scale in Eq. 609 for SOM neurons is essential for the model to explain the data. I may have missed it but I could not find a physiological justification for the value - 100msec - that is assumed for this parameter.”

We thank the reviewer for pointing out this missing information. The use of such a filter was motivated by reports on the slow recruitment of SOM neurons by sensory stimuli compared to pyramidal cells or parvalbumin cells (Yu et al., *Neuron* 104, 412; Li et al., *Cerebral Cortex* 25, 1782), which could be due to either their membrane properties or facilitating excitatory synapses onto them. We find the chosen value of 100 ms for the time constant of this equation to be physiologically reasonable. We have edited the Method section to include these references in support of the slow timescale used for the SOM neurons:

“These inputs are fed through a temporal filter of the form

$$\tau_\alpha^r \frac{dr_\alpha}{dt} = A_\alpha - r_\alpha$$

which is used to account for additional, population specific synaptic and membrane

properties in the conversion from the activity of inputs to firing rate. Motivated by experimental results showing the slow recruitment of SOM neurons (Yu et al., *Neuron* 104, 412; Li et al., *Cerebral Cortex* 25, 1782), for simplicity, we only consider $\tau_S^r \neq 0$, while taking the other two filters to be instantaneous.”

“4-On the same token, I could not find a justification for the values chosen for the rate time constant of $[\tau_E, \tau_P, \tau_S] = [10, 20, 20]$ (ms)”

While we have chosen to interpret the equations as phenomenological, with the non-linearity appearing inside of the summation, it would be appropriate to view the associated time constants as membrane time constants. To better reflect this interpretation, we have updated these parameter values to be

$$[\tau_E^m, \tau_P^m, \tau_S^m] = [10, 10, 10] \text{ (ms)}$$

as used in Litwin-Kumar, Rosenbaum, and Doiron (Litwin-Kumar, A., Rosenbaum, R. & Doiron, B. Inhibitory stabilization and visual coding in cortical circuits with multiple interneuron subtypes. *J. Neurophysiol.* **115**, 1399–1409, 2016). All figures were remade with this new parameter set, with no major deviations noted from the results presented in our original manuscript.

“5-Why considering an extended 1-D model with many columns ? Is it really necessary for the argument? It seems to me that a model involving only two columns would be sufficient. If the authors agree with that, I suggest to focus on such a model in the paper and to move the results of the full model in Supplementary data.”

We agree with the reviewer that the key result pertaining to DSI, namely that there is a transition from positive DSI to negative DSI as one moves across the tonotopic axis could be explained with a more discrete, columnar model. Unfortunately, even in a simplified case of a two-column model (a column for low and high frequency preferring neurons), we would still contend with eight differential equations, with little hope of gaining additional insight/intuition over the current model. Meanwhile, this simplification would come at the cost of ignoring the experimental data collected over the full tonotopic axis and prevent the smooth DSI transition illustrated in Fig. 4f as well as the examination of the effects of SOM cell spatial scale in Fig. 5c.

In the updated manuscript, we also make better use of our continuum model, due to a request from Reviewer #3. Namely, we present spectrally-restricted FM sweeps across the tonotopic axis on the model, similar to the experiment done in Fig. 6, and find that it is able to capture similar results. This result is now included in Supplementary Fig. 9c.

“6-The ISN regime: The authors write (line 247-252) “ Lastly, we investigate how a shift from an ISN to non-ISN regime, a change potentially resembling a shift from awake to anesthetized states (see Discussion), affects these results [...]. We find that weakening of recurrent connectivity is

enough to push the system to operate in a linear regime and eliminate both network suppression and direction selectivity (Fig.5d). Then in line 286-287 "In this study, we further demonstrate that network suppression in ISNs plays a critical role in generating direction selectivity to ethologically meaningful slow FM sweeps." I do not find the argument that the network must an ISN compelling since the authors present their results for one set of parameters. As a matter of fact, my intuition of the mechanism does not require ISN but only slow time constants and a non-linearity. The authors should elaborate more on their argument for ISN."

We agree with the reviewer that being in the ISN regime is neither sufficient nor necessary for the direction selectivity. As we proved in the Supplementary Text, which the reviewer picked up on, the crucial component is the threshold non-linearity. What Fig. 5D illustrates is that one is more likely to interact with the threshold when the circuit has strong recurrent connections that allow larger deviations from baseline activity. Nevertheless, it is possible to find narrow range of parameter sets where a non-ISN regime can lead to dynamics that interact with such a nonlinearity. Therefore, we have made the following changes to the Result section to better reflect this point:

"Lastly, we investigate how a shift from an ISN to non-ISN regime, a change potentially resembling a shift from awake to anesthetized states (see Discussion), affects these results. This is done by weakening the effective recurrent interactions between pyramidal neurons and decreasing spontaneous firing rate to 0.1 Hz. We find that weakening of recurrent connectivity is enough to push the system to operate in a linear regime and eliminate both network suppression and direction selectivity (Fig. 5d). While it would be tempting to thus conclude that ISN dynamics are a necessary condition for direction selectivity, this is not yet evident. Our result illustrates that the circuit is more likely to reach the non-linearity threshold when it has strong recurrent interactions that allow a large dynamic range of firing rates through signal amplification. Although building a robust circuit with strong recurrence likely calls for an ISN regime (Ozeki et al., *Neuron* 62, 578; Tsodyks et al., *J Neurosci* 17, 4382), it is possible that a carefully tuned non-ISN network could reach the nonlinearity threshold, albeit over a restricted range of stimulus parameters.

Taken together, our simulation results provide a theoretical foundation for nonlinear amplification of inputs in A1 and shows three circuit components that contribute to the generation of FM direction selectivity: a nonlinear neuronal transfer function, spatially broad connectivity of SOM cells, and strong recurrent connectivity."

Reviewer #2:

We are glad this reviewer found that “the question is obviously interesting and relevant, technically demanding experiments were performed and their main conclusion that this particular tuning property is calculated at the cortical level is probably right.” Below please find our responses, which address the general issues and specific comments.

Main concern:

“The authors claim that they “reveal a major contribution of recurrent network dynamics in shaping cortical tuning” (namely direction selectivity). In Figure 1h, they summarize their recordings by plotting a direction selectivity index (DSI, bound from -1 to 1) over the BF of the recorded cells. These indices are hugely variable, independent of the BF. The data shown in 1d suggests that this heterogeneity reflects biology and is not just ‘noise’. When regressing the data, an anticorrelation between DSI and BF becomes apparent. Indeed, the data looks as if there is a bias on the neurons’ direction selectivity depending on where they are positioned on the tonotopic map, perhaps hinting at some ‘edge effect’.

Importantly, however, this effect is only very small. If one would subtract this global bias from the data in Figure 1h (or would subtract the prediction from the model as shown in Figure 4f), the plot would essentially still look the same and only a minor fraction of the variability that is observed at a given BF-bin is explained in the end. However, the rest of this manuscript essentially deals with this small global effect. It does not provide insights how the pronounced and highly heterogeneous tuning properties of a set of neurons from the same position along the tonotopic axis in the end emerges. Also, the proposed network model does not capture this heterogeneity and solely reproduces the gradient in overall tuning (Figure 4f).”

To address this reviewer’s concern, we respectfully wish to point out that data from experiments in awake animals are inherently noisy due to the large trial variability caused by ongoing movements (Stringer et al., *Science* 364, 255) and arousal level (McGinley et al., *Neuron* 87, 1143). Our data ($r = 0.30$) is not especially variable even compared to the literature using anesthetized animals; for example, a recent *Nature* paper on DSI in the visual cortex (Lien and Scanziani, *Nature* 558, 80) and a *Nature Communications* paper on DSI in the auditory cortex (Sollini et al., *Nat Commun* 9, 2084) both drew conclusions with $r = 0.3-0.4$, sometimes even after exclusion of outliers. Since trial number per stimulus needs to be limited in order to avoid sensory adaptation in awake animals (Kato et al., *Neuron* 76, 962; Kato et al., *Neuron* 88, 1027), trial variability in sensory-evoked responses inevitably affects the data.

In our revised manuscript, in order to reduce the contamination from noise and represent the biologically meaningful global trend, we updated calcium imaging data by new analyses in which we raised the threshold for inclusion of cells in the analyses; cells need to be significantly responsive to at least three sounds in order to be included in data—this substantially reduces the contamination from spontaneous “significant responses” that arise from random fluctuations while largely keeping real responses, since neurons with robust responses usually respond to multiple sounds. This threshold

is described in the Methods section, and Figures 1f, 1h, 6d, and 6e are updated with this new criterion. We also included data with an even higher threshold in new Supplementary Fig. 2 as well. As you see in our new analyses, higher thresholds largely reduced the variability (Fig. 1h: $R = -0.403$, $p = 5 \times 10^{-5}$; Supplementary Fig. 2b: $R = -0.529$, $p = 8 \times 10^{-8}$). These results support the idea that the variability in our data is indeed due to random fluctuations, and if we only consider the cells with robust responses, the global effect becomes apparent. Of course, this is at the cost of becoming too selective, and therefore we present data with a modest threshold in the main figures.

This reviewer pointed to Fig. 1d as evidence against the contribution of noise, but this is not correct, since the three cells shown in Fig. 1d are from multiple fields of view, including both high- and low-A1 of the same mouse. We intentionally showed cells with positive, negative, and zero DSI as representative examples, and it has nothing to do with the variability in DSI vs BF.

We would like to also point out that the conclusion in Fig. 1h is nicely coherent with our original Supplementary Fig. 3 (now Supplementary Fig. 5) that showed the relationship between BF and DSI of synaptic currents, further solidifying our conclusion.

“When focusing on global effects in direction tuning biases, the experimental design, however, is complicated, as the hearing sensitivity in mice is most sensitive around 8-16kHz and decreases towards the edges of the hearing range. When delivering sweeps that were calibrated to maintain a constant intensity, they will nevertheless, due to the different sensitivities of the auditory system, provide an input to the cortex that changes frequency and intensity at the same time. However, it has been previously demonstrated that sounds with the same spectral content, but increasing or decreasing intensity envelope, also induce highly non-linear and asymmetric responses in the mouse auditory cortex (Deneux et al. 2016, Nat. Comm.). This confound, and how it could relate to the observations, is not mentioned at all in the manuscript.”

This reviewer is referring to Deneux et al., *Nat Commun.* 7, 12682. This study used calcium imaging in awake mice and compared A1 responses between sounds with upward vs downward intensity ramps, but with fixed spectral content. They found that A1 neurons are more strongly driven by stimuli with an upward intensity ramp, suggesting nonlinear computation in A1 or somewhere upstream. This reviewer is suggesting that the 8-16kHz preference of the mouse cochlea generates an effective ‘upward intensity ramp’ for FM sweeps moving toward 8-16kHz, which could have contributed to FM direction selectivity. There are two main reasons for why our findings are unlikely to reflect the same phenomenon they described. First, Deneux et al. used sound stimuli with fixed spectral content (white noise or 8 kHz harmonics). This means that the sound stimuli activate the same tonotopic region, and likely the same neurons, throughout the course of the intensity ramp. This is in clear contrast to our frequency sweeps experiments, in which the center of activation gradually moves across A1 tonotopy as the sound frequency changes within each stimulus, which makes it hard to think that all computation mechanisms are shared between these two phenomena. Moreover, the upward intensity ramp preference fails to account for our spectrally-

restricted FM sweeps experiments (Fig. 6). We found that FM direction selectivity is determined by the relative position between individual neuron's BF and the sweep frequency range, and not by the absolute location of the neuron's BF within the entire tonotopy. This result is inconsistent with the explanation that the overall preference of the mouse auditory system to 8-16 kHz causes direction selectivity. Alternatively, short FM results might be explainable if we assume that the upward intensity ramp preference is calculated within the receptive field of each neuron, but this hypothesis then fails to explain our long FM sweeps results (Fig. 1). Taken together, we think it best to refrain from trying to fit our results into the context of upward intensity ramp preference. Of course, even though the intensity ramp preference is unlikely to account for the global effects that we found, this does not rule out its potential modulation of direction selectivity. Therefore, to address this reviewer's concern, we have added the Deneux et al. 2017 citation as another potential source of nonlinearity and included their finding in the Discussion:

"Direction selectivity at these conditions could be partially attributed to other mechanisms, such as delay-and-compare between ON and OFF responses, combination-sensitive supralinear summation, intensity ramp-selective firing (Deneux et al. 2017), and inheritance from upstream structures."

"Additional points (major):

1) In Figure 3g, the authors present the change in DSI of recorded units with and without manipulation of SOM+ and PV+ cells, respectively. They categorize the recorded units into sweep responsive, and direction selective units. While it seems that every direction selective unit should by construction also be sweep responsive, it is difficult to interpret the data as presented. Specifically, in control conditions without light, why do some 'direction selective' units (red) have DSI indices of ~0? Related, why do sweep responsive, but 'non direction selective', units (gray) have DSI indices of ~1?"

We thank the reviewer for pointing out this potentially confusing data representation, and we regret that we did not describe the procedure for determining "significantly direction-selective" units in our original manuscript. Here, we compared the absolute DSI calculated from the real data against the absolute DSI calculated from randomized data where response amplitudes for upward and downward sweeps trials were shuffled. We determined a unit as "significantly direction-selective" when the observed absolute DSI is above 80th percentile of the shuffled data. The reason why we took this strategy is again related to the trial variability associated with recordings in awake animals. When the activity of a unit severely fluctuates across trials, this trial variability can generate a false "direction selectivity" which is determined by whether the large outlier trial happened to be in upward-sweeps trials or downward-sweeps trials. This false direction selectivity persists even after shuffling the trials between upward and downward sweeps; therefore, by only taking the units whose observed direction selectivity exceeds the shuffled data, we can exclude the noise generated by outliers. Units can have significant but small absolute DSI, if one direction reliably evoked slightly larger responses than the other direction. Similarly, units can show non-significant

“absolute DSI = 1”, if only a small number of trials evoked responses and all the other trials (including the same and the opposite direction) evoked nothing. The procedures for determining the significant direction selectivity is now described in the Methods section under “Analysis of single-unit recording data”:

“To avoid the false direction selectivity which arises from random trial variability, significance of direction selectivity was determined by comparing the absolute DSI calculated from the real data against those calculated from randomized data where response amplitudes for upward and downward sweeps trials were shuffled (1000 repetition). We determined a unit as “significantly direction-selective” when the real absolute DSI is above 80th percentile of the shuffled data.”

We would like to also note that our SOM cell inactivation-specific reduction in direction selectivity holds true even if we include both significantly direction-selective and nonselective units, demonstrating the robustness of our conclusion (only significantly direction-selective: SOM, $p = 0.0001$; PV, $p = 0.336$; both direction-selective and non-selective: SOM, $p = 0.0006$; PV, $p = 0.667$).

“2) The authors cite a number of studies well known in the field suggesting a sub-cortical origin for direction tuning. They claim that the contradicting observations in the current study were due to different experimental conditions, i.e. anaesthetized vs awake mice. However, to support this important claim, and to resolve this apparent contradiction, it would crucial to compare these conditions side-by-side.”

Here, this reviewer’s comment touches important points that can be separable to two questions: namely, cortical vs subcortical origins of direction selectivity and awake vs anesthetized experimental conditions. Therefore, we address these two questions separately below.

First, we thank the reviewer for bringing the concern regarding cortical vs subcortical origins to our attention. Actually, we did not have any intention to claim that direction selectivity is generated solely in the auditory cortex, and that is why we mostly used the word “shape” in our original manuscript (e.g. the title “Recurrent network dynamics shape direction selectivity in primary auditory cortex”). There is no doubt that direction selectivity is calculated upstream of the auditory cortex, as you see in our citations, and we are proposing that the computation within A1 circuits further shapes it. Therefore, we do not see our results as contradictory to previous literature which showed direction selectivity in subcortical structures. To express this idea we stated in our original Discussion that “Direction selectivity at these conditions could be partially attributed to other mechanisms, such as ... inheritance from upstream structures”, but we agree that the wording might not have been clear enough and it gave the wrong impression. To make this point more clear, we changed the wording in Result section and also added an entire new section in Discussion to discuss potential subcortical origins of direction selectivity:

“Cortical and subcortical sources of FM direction selectivity

FM direction-selective firing has been previously observed in subcortical auditory

structures, including cochlear nucleus, inferior colliculus, and thalamus. In particular, direction selectivity in the inferior colliculus has been extensively studied in bats for its relevance in their echolocation and communication, and its generation is considered to involve temporal offsets between feedforward synaptic inputs. Therefore, although we demonstrated SOM cell-mediated cortical enhancement of FM direction selectivity, it is likely that a part of direction selectivity in A1 is inherited from upstream peripheral structures. This idea is consistent with residual direction selectivity we observed after SOM cell inactivation (Fig. 3g), and both inheritance from upstream and recurrent circuit-mediated enhancement are compatible with asymmetry in FM sweeps-evoked postsynaptic charges (Fig. 4). In fact, assigning a moderate FM direction selectivity to the feedforward excitatory input in our network model supported the additive nature of direction selectivity from cortical and subcortical mechanisms (Supplementary Fig. 9d-g).

How neuronal computations in the sensory periphery contribute to cortical information processing has been debated across sensory modalities. Similar to our findings in the auditory system, generation of direction selectivity in the visual system has been reported both in the cortex and peripheral retina. Ablation of direction-selective retinal ganglion neurons in a previous study only partially reduced direction selectivity in the primary visual cortex, demonstrating additive contributions from both peripheral and cortical mechanisms. Interestingly, this retinal perturbation affected cortical responses mostly at high-speed visual motion (40°/s), suggesting that cortical mechanisms enhance visual motion processing predominantly at low- to mid-speed. Likewise, we found in our network model that the enhancement of FM direction selectivity within the A1 circuit is less prominent at high-speed FM sweeps (Fig. 4g). Thus, complementary extraction of rapid and slow time-varying stimuli in the periphery and the cortex—which likely relates to the loss of faithful encoding of rapidly varying stimuli at the cortex level—may be a general feature shared across sensory modalities. Elucidating how the cortex inherits and transforms various sensory information from the periphery and expands encoding capacity will be a critical step in understanding cortical computations.”

Instead, what we see as a real contradiction from our results is the lack of direction selectivity at slow FM rates in the A1 of anesthetized rodents (Zhang et al., *Nature* 424, 201), as we explicitly stated in our original manuscript. To examine if this controversy is really due to the difference between the awake and anesthetized state, we performed additional single-unit recording experiments to compare direction selectivity between brain states. Unexpectedly, we observed a quite different result from Zhang et al., who reported a complete lack of direction selectivity below 30 oct/s; under urethane anesthesia, we found that regular-spiking neurons were direction-selective across the entire range of FM rates, and if anything, DSI was slightly higher at slow FM rates (see the figure below for a direct comparison). Another recent study that we cited in our manuscript (Sollini et al., *Nat. Commun.* 9, 2084) also reported robust direction selectivity of A1 neurons to slow FM sweeps under medetomidine anesthesia, and we are not sure why Zhang et al. did not see this. A paper by Wehr and Zador (*Neuron* 47, 437) previously reported an inconsistency between their results and Zhang et al. regarding kinetics of synaptic currents and attributed it to the use of pentobarbital anesthesia used in Zhang et al. Since pentobarbital enhances GABA transmission and

likely changes computation in both cortical and subcortical structures, it is possible that the difference we observed might be also due to their use of pentobarbital. Unfortunately, we do not have permission to use pentobarbital in our lab, and comparing the difference between anesthetics is beyond the scope of this study.

Nevertheless, we do want to emphasize that our new data clearly shows an enhanced DSI in awake compared to anesthetized state, and the range of FM rate with DSI enhancement is consistent with our model. Therefore, we include this result as new Supplementary Fig. 3 and cited this finding in Result and Discussion, respectively, as:

“Nevertheless, the absolute DSI value remained high regardless of FM rate (Fig. 1f and Supplementary Fig. 2); we also confirmed this result with single-unit recordings and found enhanced direction selectivity for slow FM sweeps in the awake state compared to the anesthetized state (Supplementary Fig. 3).”

“Furthermore, the range of FM rates where we observed enhanced direction selectivity in the awake state compared to the anesthetized state was consistent with our model (Fig. 4g and Supplementary Fig. 3). “

“3) In optogenetic experiments a prime parameter is the intensity of the light that is being applied, as the observed effects may critically depend on it. Here, the authors normalize the inhibition by PV+ and SOM+ cells by normalizing to spontaneous output firing of excitatory neurons (Figure 3h). It seems unclear why 1.5-fold increased spontaneous firing is used for normalization. The effect on the spontaneous activity levels may depend on qualitative differences between PV+ and SOM+ cells as they may have a qualitatively different influence on direction selectivity (as the authors report). However, this is an arbitrary approach. To illustrate this, just consider the inverse, such that the light intensities would have been first adjusted such that PV+ and SOM+ cells would have a similar effect on direction tuning, and then the differential effects on spontaneous firing rates would have been described. Demonstrating the robustness of the effects across a wide range of light intensities would help convincing the reader that the effects are indeed dependent on the targeted cell-type and not on the specific light intensity settings.”

We thank the reviewer for pointing this out. To address this reviewer’s concern, we performed an additional analysis comparing the change in spontaneous firing rate

against change in absolute DSI of individual units and included the results in Supplementary Fig. 7b. This new data shows that the reduction of DSI in SOM-eNpHR3 experiments and slight increase of DSI in PV-eNpHR3 experiments hold true in units with a range of spontaneous firing change. This result supports that our conclusion is not an artifact due to the differential level of effects of PV cells and SOM cells on spontaneous firing rates.

We would like to also note that it is not possible to adjust light intensities “such that PV+ and SOM+ cells would have a similar effect on direction tuning”, since we observed opposite directions of changes by inactivation of SOM cells (significant reduction in DSI) and PV cells (tendency for enhancement in DSI). This is consistent with the previous finding that inactivation of SOM cells reduces network suppression whereas inactivation of PV cells slightly enhances it (Kato et al., *Neuron* 95, 412). Furthermore, we selected our light intensity criteria of “not to exceed 100% increase in population spontaneous firing” based on our previous observations using both unit recordings and whole-cell recordings that the excessive inactivation of inhibitory neurons beyond this point paradoxically eliminates sound-triggered responses and often irreversibly changes the cortical activity to a bursty state. This elimination of sound-evoked activity is likely due to the saturation of cortical firing at a high activity level (Rubin et al., *Neuron* 85, 402). Both changes in activity state and elimination of sound-evoked activity make it impossible to assess the cortical computation. We made this clear by rewriting the Methods section to state:

“Since we found that excessive inactivation of inhibitory neurons causes a paradoxical elimination of sound-triggered responses and often irreversibly changes the cortical activity to a bursty state, LED intensity was kept at the lowest effective intensity (1-3 mW/mm² at the brain surface),”

and referred to this in the Results section as:

“We used weak LED intensities to modulate transmission without inducing irreversible changes in the cortical activity states (see Methods).”

“Additional points (minor):

1) *It is unclear whether the data presented in Figure 1h stems from all tested rates or not. Considering the importance of rate in many other quantifications, this is confusing.”*

The data in Figure 1h includes all tested rates, and this is now stated in the figure legend as:

“DSI of pyramidal cells averaged across all FM rates have a strong dependence on their BF.”

“2) *The title of Figure 2 highlights the asymmetry of excitatory inputs between FM directions, which the presented data clearly supports. But why do the authors focus on excitation, when*

inhibitory inputs seem to be equally asymmetrically tuned (Figure 2d left)?”

The title was revised to:

“Direction selectivity is caused by the asymmetry of sound-evoked postsynaptic charge between FM directions.”

“3) In the model, there seems to be a strong dependence of absolute rates of FM sweeps on DSI (Figure 4g). This is not the case for the experimental data (Figure 1f or Figure 2e).”

All our updated data, including Fig. 1f, Fig. 2e, Fig. 6d, and new Supplementary Fig. 2 and Supplementary Fig. 3 show a dependence of absolute DSI on FM sweep rates. Although it is likely that the contribution of noise in the real brain makes the dependence less prominent compared to the theoretical model, all our experimental data are coherent with our modeling result (Fig. 4g). We changed the scale of the plots to emphasize this dependence in the updated figures.

“4) There are different meanings of the acronym TRF, it should be defined in the manuscript.”

Thank you for pointing this out. We must have accidentally dropped the definition during editing. We now spelled out the full words in the Results section.

“5) Throughout the text statistical p-values are given, however, no further information on what exactly was tested, e.g. what test was being applied.”

In our original manuscript, we had a “Statistical analysis” subsection under Methods, stating that “two-tailed paired t test was used, unless otherwise stated.” To make our results more transparent, we now include Supplementary Table 4 which includes the statistical analysis for each figure.

“6) Figure 5a illustrates the effects on direction tuning when simulating the optogenetic manipulations in the network model. Here, the middle panel showing the firing rate in response to upward or downward sweeps still reveals a substantial asymmetry (alike Fig. 4dv), although the claim would be that this difference would vanish upon the block of SOM+ cells.”

It might appear that there is still asymmetry in Fig. 5a, but the panels on the left and right are the accurate quantification of the results, which show the disappearance of asymmetry. Please note that there is also a difference in the amount of suppression below baseline between upward and downward sweeps. This is significant because the definition of DSI compares the area under these two curves. While the curves appear different in shape, the areas under them are very similar.

Reviewer #3:

We are delighted that this reviewer found that our experiments “provide compelling evidence”, and we also appreciate the thought that “Overall, this is a really beautiful study. The data address the narrow question of how FM direction is encoded in the mouse auditory cortex but also extends to bigger issues.” Below are the responses to this reviewer.

“I see only one substantial issue with their manuscript that is easily fixed by text changes and possibly a few additional changes to their model.

General points:

1. They over-interpret their data when they write, “FM direction selectivity emerges in cortical circuits” (ln 201), “Our SOM cell photoinactivation experiments demonstrated the generation of direction selectivity to slow FM sweeps within the cortex (ln 335-6), “revealing its cortical origin” (ln 33). Subcortical neurons clearly have FM direction selectivity has evidenced by dozens of papers documenting the phenomenon in the inferior colliculus and medial geniculate body (and in the brainstem as well). FM direction selectivity definitely DOES NOT originate in the cortex and I think choice of wording will really hit a nerve with readers. What they data do show – very elegantly and convincingly – is that SOM cells greatly enhance / strongly shape etc. FM selectivity at the level of the cortex. This can be as would like to see an expanded section of the Discussion that addresses the point that FM direction selectivity is already constructed subcortically. As it stands, they cite reference 8, which did not even record from a subcortical structure. For example, it would be important to consider their findings in relation to George Pollack’s work on FM direction coding in the inferior colliculus (yes, it is a bat but it’s not an echolocating bat, so it’s basically the same as a mouse).

Their over-interpretation comes from a combination of not citing the subcortical literature but also over-interpreting the lack of fast EPSC/IPSC for non-preferred directions. 1) Their assertion that thalamic inputs have no direction selectivity is based on recordings from L2/3 pyramidal neurons and as such is not the basis for ruling out what the cortex can receive from the thalamus, 2) the fast EPSC/IPSC does not definitely isolate a thalamocortical contribution, 3) even their favorite mechanisms, recurrent activity, can depend critically on ongoing input from the thalamus (see Reinhold and Scanziani) and they have not ruled out that thalamic inputs that either target or disproportionately drive SOM neurons is not critical for findings they report here. In future work, they could use 2-p imaging of MGB thalamocortical terminals or silence cortical activity pharmacologically or with PV-ChR2 while recording EPSCs across middle layers to isolate what the MGB is transmitting to the cortex.

Apart from backing off of their wording and adding citations to the Discussion, I am concerned that their model parameters are not set realistically because they assume the DSI for thalamic inputs is zero. Could a supplemental figure address how slight increases in thalamic DSI would affect their findings?

My only motivation here is to help the authors showcase their work in a way that will resonate

with the community of researchers that work on the auditory system. Clearly, I am a fan of this paper and admire the work that this lab is publishing. If they back off the language that direction coding originates in the cortex, do a better job of citing the literature demonstrates direction coding subcortically, and include a portion of the Discussion that considers the cortical circuit mechanism to what is constructed subcortically and what is passed on through thalamocortical synapses (the two are not the same), I think this beautiful paper will get the higher citation counts that it deserves.

We are grateful to this reviewer for the constructive suggestions for improving our manuscript. Actually we did not have any intention to claim that the direction selectivity is generated solely in the auditory cortex, and that is why we mostly used the word “shape” in our original manuscript (e.g. the title “Recurrent network dynamics shape direction selectivity in primary auditory cortex”). However, we admit that our wording was not clear enough and it gave the wrong impression. There is no doubt that direction selectivity is calculated upstream of the auditory cortex, and what we wanted to propose was that the recurrent circuits in the auditory cortex further enhances it, exactly as the reviewer described. To make this point more clear, we followed the suggestion of this reviewer by changing the wording in Result section and adding a section titled “Cortical and subcortical sources of FM direction selectivity” in Discussion:

“Cortical and subcortical sources of FM direction selectivity

FM direction-selective firing has been previously observed in subcortical auditory structures, including cochlear nucleus, inferior colliculus, and thalamus. In particular, direction selectivity in the inferior colliculus has been extensively studied in bats for its relevance in their echolocation and communication, and its generation is considered to involve temporal offsets between feedforward synaptic inputs. Therefore, although we demonstrated SOM cell-mediated cortical enhancement of FM direction selectivity, it is likely that a part of direction selectivity in A1 is inherited from upstream peripheral structures. This idea is consistent with residual direction selectivity we observed after SOM cell inactivation (Fig. 3g), and both inheritance from upstream and recurrent circuit-mediated enhancement are compatible with asymmetry in FM sweeps-evoked postsynaptic charges (Fig. 4). In fact, assigning a moderate FM direction selectivity to the feedforward excitatory input in our network model supported the additive nature of direction selectivity from cortical and subcortical mechanisms (Supplementary Fig. 9d-g).

How neuronal computations in the sensory periphery contribute to cortical information processing has been debated across sensory modalities. Similar to our findings in the auditory system, generation of direction selectivity in the visual system has been reported both in the cortex and peripheral retina. Ablation of direction-selective retinal ganglion neurons in a previous study only partially reduced direction selectivity in the primary visual cortex, demonstrating additive contributions from both peripheral and cortical mechanisms. Interestingly, this retinal perturbation affected cortical responses mostly at high-speed visual motion (40°/s), suggesting that cortical mechanisms enhance visual motion processing predominantly at low- to mid-speed. Likewise, we found in our network model that the enhancement of FM direction selectivity within the A1 circuit is

less prominent at high-speed FM sweeps (Fig. 4g). Thus, complementary extraction of rapid and slow time-varying stimuli in the periphery and the cortex—which likely relates to the loss of faithful encoding of rapidly varying stimuli at the cortex level—may be a general feature shared across sensory modalities. Elucidating how the cortex inherits and transforms various sensory information from the periphery and expands encoding capacity will be a critical step in understanding cortical computations.”

We also constructed a new network model which incorporates a small direction selectivity in the inputs from the thalamus (new Supplementary Fig. 9d-g). We have implemented thalamic DSI in the following way (these details are now included in the Methods section):

$$Amp_{\alpha}(t) = \begin{cases} 0, & \text{for } t < t_{start} \text{ and } t > t_{stop} \\ Amp_{max} + \frac{Amp_{min} - Amp_{max}}{t_{stop} - t_{start}}(t - t_{start}) & \text{otherwise} \end{cases}$$

where Amp_{min} is $0.8 \cdot Amp_{max}$, and t_{start} and t_{stop} are the start and stop times of stimulus, respectively.

This adjustment decreases the amplitude of the feedforward stimulus by 20% in a linear fashion from start to finish. We reproduced Fig. 5 with this model, keeping all other model parameter values the same, and found overall the same conclusions. However, we do note that the intuition of the reviewer was correct: the direction selectivity generated by thalamic and cortical mechanisms were additive, and thus DSI in this new model was overall slightly higher than the original model. While blocking SOM cells still significantly attenuated direction selectivity, there is remaining DSI inherited from the thalamic input. We included the following statement in the new “Cortical and subcortical sources of FM direction selectivity” section of the Discussion:

“In fact, assigning a moderate FM direction selectivity to the feedforward excitatory input in our network model supported the additive nature of direction selectivity from cortical and subcortical mechanisms (Supplementary Fig. 9d-g).”

“2. For the optogenetics experiments, they should clarify what type of units they are recording from. Presumably, they could sort their sample into regular spiking and fast spiking based on the spike waveform. Further, are they able to identify SOM neurons based on spiking changes arising directly from activating the opsin?”

Either way, to make the single unit recordings more comparable to the model they could focus only on the RS putative pyramidal units. If they have enough FS/PV units, they could provide a supplemental figure to show how SOM inactivation affects PV DSI and how this is predicted by the model. Obviously, if they have enough optogenetically identified SOM neurons, it would be hugely important and interesting to show that they have the broad FM selectivity assumed by the model (e.g., in Fig. 4d, iv).

We thank the reviewer for suggesting this important analysis. We have performed additional analyses to separate fast-spiking units from regular-spiking units. Now our new Fig. 3g and 3h include only regular-spiking units, and the results stayed the same. Although the number of fast-spiking units is not enough to draw clear conclusions, in the revised manuscript we included the data as Supplementary Fig. 7a for full transparency.

We also tried identifying SOM cells as rapidly photoinactivated units during the SOM-eNpHR3 experiments but found only two single-units that we feel confident in their rapid inactivation (shown in the figure below)—we did find other units which showed either slow or modest photoinactivation, but we did not feel confident enough regarding their identity.

As you see, SOM unit #1 showed a predicted pattern of firing: firing at a wide range of frequencies during FM sweeps, with only small direction selectivity. In contrast, SOM unit #2 showed sweep-triggered suppression of firing and direction-selectivity was unmeasurable.

It is likely that there are more SOM+ units in our recordings. However, identifying them is not straightforward for two reasons; first, we are only partially inactivating their activity. Furthermore, as predicted for inhibition-stabilized network, optogenetic inactivation of inhibitory neurons can paradoxically increase their firing, which also makes it challenging to distinguish SOM units from others. (You can also see from our fast-spiking units data that they are still actively firing during PV cell photoinactivation.) Since we have only two SOM+ units, we decided not to show them separately in our revised manuscript.

It would be helpful if some of the points above could be addressed in a supplemental figure alongside more examples of single units recorded in the two optogenetic silencing conditions. The two examples in Fig. 3 have very different firing rates and it is unclear why the evoked responses occurs at the same time for Pref and Non-Pref in 3C unless this cell is tuned to ~16kHz or has much broader frequency tuning. For this reason, it would be helpful if they could indicate with an arrow in all of their FM-evoked activity traces when the sweep crosses through the BF/TRF of the neuron or – better yet – include the tonal receptive field for each of the units they show in this figure and elsewhere.”

This is a good point, and exactly as this reviewer predicted, the unit shown in Fig. 3C is tuned to 16 kHz (tuning properties are now included in our new Supplementary Fig. 6). It might look inconsistent with our model that a 16kHz-preferring neuron shows an upward direction selectivity; however, this is likely because the receptive fields of neurons are not perfectly symmetrical even in the mid-frequency area of A1, and the intensity of network suppression tends to be stronger on the high-frequency side (see Kato et al., *Neuron* 95, 412, Fig. 1 and Fig. 4). Consistent with this, Fig. 1h of our manuscript shows more positive DSI at 16 kHz BF, which is slightly off our model. The reason why tonal receptive fields are skewed is not clear—maybe it stems from the skew existing at the cochlea or asymmetry of connectivity within A1—and it is hard to incorporate this skew into our current model, but this is something we could potentially improve in the future network models.

Following this reviewer's suggestion, we now included more single-unit examples in our Supplementary Fig. 6 together with their pure tone tuning properties, measured by presenting tones with nine frequencies at 70 dB SPL.

"3. Could they clarify whether all recordings were made in the left hemisphere, as suggested in Fig. 1? Hemispheric differences in FM sweep coding have been reported previously and based on this literature the right hemisphere would be expected to show stronger direction coding."

We apologize for the confusion. All our recordings were performed in the right hemisphere, as we wrote in the Methods section of our original manuscript, but the drawings in the figures were not showing it. In the revised manuscript, we flipped the speaker positions of the figures to reflect the actual experimental setup.

"4. It would be helpful if the authors provided a cartoon schematic for the prevailing model (FM direction created by fast IPSC timing) and the SOM/ISN model proposed here. It would be nice if this were included at the end of Fig. 1 or the beginning of Fig. 2 to put the data into context."

Thank you for the suggestion. We now included a new schematic as Fig. 2g, just before the data where we examine the temporal offset hypothesis.

"5. Could the ISN model also capture the attenuation of fast IPSCs during the non-preferred direction of FM sweeps?"

The model also captures the attenuation of fast IPSCs at different wave speeds. This result is added to our manuscript as Supplementary Fig. 9a and we adjusted the following sentence in the text.

"This network suppression silences the recurrent activity and leads to an attenuated amplification of both excitatory and inhibitory inputs (Fig. 4e and Supplementary Fig. 9a)."

“6. What are the temporal offsets of the EPSCs and IPSCs from the ISN model? Does it also show a comparable timing of EPSCs and IPSCs?”

Similar to the experimental data, there was no difference in either the EPSC-IPSC peak time intervals or rise time intervals between the preferred and non-preferred directions, and therefore timing difference is not a source of the observed DSI. This result is now included in new Supplementary Fig. 9b.

We note that we observed slight leading EPSCs in the peak time, which is attributed to the phenomenological model's time constant $\tau_m = 10$ that imposes a lag to all recurrent inputs compared to the feedforward input. We also observed slight leading IPSCs in the rise time, which is due to the broader tonal receptive field of IPSCs compared to EPSCs (which you also find in our experimental data Fig. 2k).

“7. In the framework of network suppression by the non-preferred direction of FM sweep, the schematic (Supplementary Figure 5C) illustrates how the network suppression can recapitulate the direction of selectivity based on the central frequency of FM sweeps and BF of a neuron. Could the ISN model in Figure 4 and 5, predict the phenomenon in Figure 6?”

We thank the reviewer for suggesting that we investigate this phenomenon with our model. Similar to the experiments, we considered seven spectrally-restricted sweeps covering one octave of the tonotopy with the speed of 20 oct/s. Consistent with our experimental results, we find that the DSI of a population of cells for a given stimulus is determined by its relative location to the center of the sweep. One difference from our experimental result is that, due to the fixed bandwidth of modelled neurons' tonal receptive fields, populations too far from F_{center} experienced only network suppression and did not respond to the stimulus. Therefore, DSI in such regions become uninterpretable, hence the plot lacks the points that return to a DSI of zero. This result was included as Supplementary Fig. 9c and the following text was added to the manuscript:

“Remarkably, when DSI was calculated separately for bins of relative F_{cent} position, we found a sharp reversal of DSI at around $F_{\text{cent}} = 0$ (Fig. 6e), a result also captured in our computational model (Supplementary Fig. 9c). Therefore, a neuron shows preference to the direction in which FM sweeps move away from its BF.”

“8. Figure 4d, subpanel v shows the firing rate of a PyrN from the model for preferred and non-preferred directions. Figure 5a shows the firing rate of a PyrN with the same BF, only this time with a partial block of SOM inputs. The issue is that both of these plots look nearly identical and it is unclear why/how that would be.”

It might be hard to capture by the eyes, but a direct overlay of two plots shows that SOM cell partial blockade increases both the height and width of the excitation and reduces the suppression triggered by the downward sweep (the right panel shows the traces from our original model, but the conclusion remains the same in our new model updated in response to the Reviewer #1's comment).

“9. On Ln 259 they mention that shorter FM sweeps are prevalent in mouse vocalizations. It seems they have the acoustic recordings to substantiate this claim but the evidence is not shown. Can it be provided? Further, I suggest that they use the term “restricted” FM instead of “shorter” bcs the latter is confusing about whether they refer to time or frequency.”

We analyzed the FM frequency ranges of individual continuous segments in mouse vocalizations and added the results to Supplementary Fig. 1. Across all vocalization types and strains, the frequency ranges mostly stayed below 1 octave and never exceeded 2 octaves, consistent with our notion that FM sweeps in vocalizations are dominated by restricted frequency ranges.

We also changed the text to call them either “spectrally-restricted FM sweeps” or “FM sweeps with restricted frequency ranges.”

“10. Line 131: when first mention TRF, please include the full name of the abbreviation.”

Thanks for pointing this out. We now spelled out the full words in the Results section.

“11. Lines 139-140 – I disagree with their statement, “consistent with the idea that inhibition is recruited through recurrent excitation within A1”. Although I think the SOM inhibition data they present later supports this view, at this point in the manuscript the findings they observe could be explained equally well by DSI that is entirely inherited from upstream neurons that have direction selectivity. In either case, you would have no EPSC/IPSC for the non-preferred direction. In fact, they could use this ambiguity as the motivation to perform the optogenetics experiment because it would make for a nice transition.”

Thank you for the suggestion. We removed the statement and changed the original lines 139-140 to:

“Similar to fast EPSCs, we observed a strong attenuation of fast IPSCs in non-preferred FM sweeps, and DSI_{IPSC} followed the same trend as DSI_{EPSC} .”

The beginning of the optogenetics section was already written in a way that presents the two hypotheses, inheritance from thalamus vs cortical computation, so the story flows well.

“12. Line 641: what is ode45?”

ode45 is a numerical solver for differential equations that is implemented by Matlab (MATLAB, version 9.6.0 (R2019a), Natick, Massachusetts: The MathWorks Inc.; 2019) and is based on an explicit Runge-Kutta (4,5) formula, the Dormand-Prince pair. The following text was adjusted to clarify this point.

“All simulations were completed using ode45, a numerical solver for differential equations implemented by Matlab, with maximum relative tolerance and absolute error levels set to 1×10^{-5} and 1×10^{-6} , respectively.”

“13. Line 653: “The code for all simulations can be found on GitHub (https://github.com/gregoryhandy)”. The code is not yet on the GitHub site.”

We apologize for the delay in posting the code. It is now publicly available on GitHub.

“14. As a technical curiosity, does 615nm excite tdTomato? How does increased endogenous fluorescence affect the ISI quality?”

No, tdTomato excitation spectrum drops to zero at around 600 nm, and there is negligible excitation by 615 nm light. We also realized that we mistakenly wrote the wavelength as 615 nm, but the actual LED used in intrinsic signal imaging was 625 nm (corrected in the revised manuscript), which makes it even less likely that tdTomato gets excited. We have not realized any difference in the intrinsic signal imaging between wildtype mice and mice with tdTomato.

“15. Looking at the asterisks in Fig. 2b, it’s not clear how they operationally define the contribution of network suppression. How do they distinguish between sound-evoked synaptic currents that end versus sound-evoked synaptic currents that are actively suppressed? In both cases, the currents would be reduced.”

We consider the signal as network suppression only when the current goes below the baseline level. A simple termination of an excitatory current would not fall below the baseline, so it would not be considered as network suppression.

“16. The writing style for the last sentence on Pg. 6 (ln 126-129) could be improved. It’s a run-on in its current form.”

We rewrote the sentence as:

“Network suppression is a suppression of recurrent activity that is evoked by a broad range of non-preferred frequencies; therefore, sustained network suppression is

triggered as our 4-octaves FM sweeps pass through these non-preferred frequencies.”

“17. Line 167, “We used weak LED intensities to modulate transmission without inducing aberrant discharges” and Ln 487 “to prevent aberrant sound-evoked cortical activity”. They only take units where the firing rate was increased by 25-100% during illumination (489-90), which by definition is aberrant (i.e., not normal). They are changing the spike rates by inactivating PV and SOM neurons so they should not try to cover this up. Moreover it isn’t clear why they report the power as 10-35 mW. Is this at the fiber tip? Why not express as mW/mm² as measured on the brain?”

As we wrote on our comments to the Reviewer #2’s major comment 3, we selected our light intensity criteria of “not to exceed 100% increase in population spontaneous firing” based on our previous observation that the excessive inactivation of inhibitory neurons beyond this point paradoxically eliminates sound-triggered responses and often irreversibly changes the cortical activity to a bursty state. This is what we meant by “aberrant” activities, but we should have been more clear about this point. To improve this, we rewrote the Methods section to state:

“Since we found that excessive inactivation of inhibitory neurons causes a paradoxical elimination of sound-triggered responses and often irreversibly changes the cortical activity to a bursty state, LED intensity was kept at the lowest effective intensity (1-3 mW/mm² at the brain surface)”

and referred to this in the Results section as:

“We used weak LED intensities to modulate transmission without inducing irreversible changes in the cortical activity states (see Methods).”

“Lastly, how was the “lowest effective intensity” determined and can this be appended to the Methods?”

We included a description of the procedure in the Methods section:

“Before starting measurements of sound-evoked responses in each mouse, we monitored spontaneous activity without sound stimuli while applying brief LED illuminations, starting from a low intensity and incrementally with higher intensities. We determined the lowest effective intensity that caused a visible increase in spontaneous activity and used that for the FM sweeps experiments.”

“18. Line 257, pure tone FM sweeps are not ethologically meaningful. The rates of FM they use are ethologically relevant. I suggest the wording be changed.”

There were several lines in the text which used “ethologically meaningful”, so we changed all of them to “ethologically relevant.”

“19. Line 410, the preparation of the skull for ISI is not clear. Is the skull intact? Thinned? Made transparent by wetting it with saline?”

We added a sentence to the Methods stating:

“The skull was kept transparent by saturation with phosphate-buffered saline.”

“20. Line 432, How are the images and cells aligned for the two imaging sessions to confirm that pure tone and FM measurements are made from the same cells?”

We added a sentence to the “Analysis of two-photon calcium imaging data” section of Methods:

“ROIs were aligned across days using affine transformation of the ROI positions, and ROIs from two days were judged to be the same cell if there is more than 60% overlap in the areas. Furthermore, a custom graphical user interface was used to visually inspect individual ROIs for the appropriate alignment.”

“21. Line 474, why do they inject the eNpHR3 virus in newborn mice instead of young adults like the GCaMP?”

We have repeatedly observed that the overexpression of optogenetic tools under strong promoters (e.g. CAG or EF1 α) by AAV injection in adult animals causes severe cell death. We could avoid this by diluting the AAVs to a lower titer, but this is at the expense of poor spread in the cortex. We have found that injections in newborn pups give better results in maintaining cell health while achieving wide-spread expression.

Our GCaMP-expressing AAV uses a weaker synapsin promoter, and therefore its injection in the adult mice causes fewer problems (although we still need to keep the titer relatively low).

“22. Line 485, what do they mean thinned skull? They made a craniotomy so isn't the light from the LED directed at the exposed brain?”

The craniotomies we make for electrophysiology are very small (<300 μ m), and the skull surrounding the craniotomy (<1 mm) is thinned. Therefore, the light largely penetrates through the thinned skull, and not necessarily through the craniotomy. To make this point clearer, we revised the sentence to:

“A fiber-coupled LED (595 nm) was positioned 1-2 mm above the thinned skull and a small craniotomy.”

Reviewers' Comments:

Reviewer #1:

Remarks to the Author:

As I wrote in my first the results of this paper are important for our understanding of the network mechanisms underlying the processing of complex auditory stimuli. They have an even broader relevance since direction selectivity is also observed for other sensory modalities. The paper overall is well written and understandable equally well by experimentalists and theoreticians. Therefore, it fits well for the general audience of Nature Communication.

My first report raised six issues. I am fully satisfied by the changes made to address five of them.

The last one concerns the implications of the results for the dynamical regime in which the auditory cortex operates. The authors have now modified the manuscript to address this issue. However, to make their argument more clear, I would like to suggest: 1) to summarize in the first subsection of the discussion the main conclusions of work. 2) To include in this subsection the intuitive description of the mechanism without referring to the ISN since in principle the mechanism does not require the network to operate in this regime. 3) to add a subsection discussing with the role of the ISN and the interpretation of the difference between anesthetized and awake state. 4) To move lines 262-268 in this subsection.

Reviewer #2:

After reading the comments from the other reviewers, it appears that I am the only reviewer having raised serious concerns about the manuscript by Aponte et al. I therefore do not want to block this manuscript, but still must say that I do not feel that major concerns were addressed adequately in the rebuttal. On the one hand the authors argue that their data is noisy, on the other hand, they want to claim to have identified a major mechanism explaining the emergence of direction selectivity, which is difficult with noise-dominated data. I must admit, that I still could not develop an estimate how solid the data is in the end. The revisions therefore did not lead to an increase in my enthusiasm about this manuscript.

Again, my first major concern was/is: the effect of global biases in the preferred DSI is rather small compared to the large heterogeneity that they report for a given spot along the tonotopic axis. This local heterogeneity represents a much larger fraction of the overall variability in the DSIs than the bias in DSI that is seen along the tonotopic axis. The authors argue in their rebuttal that this heterogeneity is essentially due to trial to trial-noise and can therefore be disregarded. I pointed out to Figure 1d, where they claim to show three representative examples of DSI measurements. Averaged sound responses were calculated to diminish trial-to-trial variability and they obtain impressively clean and nice and gradual direction tuning curves. That these three particular examples have been recorded from different spots along the tonotopic axis was not the point I was making. The point is, that if Figure 1d is true and representative, their assessment of the DSI of a given cell is rather exact and not noise-dominated at all. When the authors then would take their own data serious, as plotted in Figure 1h, or even Figure 1g, they would acknowledge that apparently a lot of real, functional heterogeneity in DSIs exist at the local scale, which, however, is simply ignored by their proposed model:

Zoom in in Figure 1g, left homogeneity in BF, right heterogeneity in DSI

The response of the authors to my concern was essentially to apply another data selection criterion such that the analyses better fit the statements they want to make, however, even now, there is still a lot of local DSI heterogeneity. I think when their paper will be read by critical readers, they could reach a higher acceptance by at least adjusting their conclusions and to acknowledge what their data says (much of the DSI heterogeneity is local and not captured by their proposed model / mechanism) and then qualify, that they are nevertheless focusing on this global bias. This is legitimate. But just ignoring the major effect in their data will leave the reader puzzled.

The second major concern was that the sweep stimuli used in the study – at the cortical level – may confound two parameters: pitch and intensity. The author's argument in the rebuttal that their observations could not have anything to do with the findings of Deneux et al, since they used stimuli in which only one parameter is changed (namely intensity) is hard to follow for me.

I think it is an improvement, that the authors at least mention this possible confound now in the discussion section.

Another major point was the data shown in Figure 3g. Again, here, the argument of noisiness is being used to try to address my concern: I found the plot difficult to read, as there are apparently many cells that have been classified as sweep selective, but **not** direction selective (grey), that have surprisingly an DSI of 1 (grey arrows). Also, there are several cells that have been classified as both, sweep and direction selective (red), that barely show any direction selectivity as quantified with DSI (red arrows).

Figure 3g, right panel.

Sure, noisiness can lead to a surprisingly high DSI, that turns out not to be significant (caused e.g. by a single major response) and if suddenly the trial to trial variability in a given cell apparently is very low, one could observe a small DSI, that is nevertheless significant. But overall, if some statistics is applied to achieve a meaningful categorization into direction selective and non-selective cells, one would nevertheless expect some separation of the data according to the categories. Here, I just consider the x-values of the puncta and their distribution along the x-axis, describing the DSIs under basal, control conditions. If grey dots should represent **non**-direction selective cells, I would expect a relative enrichment of 'grey' cells to the left, and if the red dots should represent direction selective cells, they should be relatively enriched for higher DSI values. For the x-values smaller than 0.4, I counted 9 grey / 15 red (0.6) and for x-values larger than 0.4, I counted 4 grey / 6 red (0.66), essentially a similar ratio. Admittedly, if one would have made this calculation with a threshold of DSIs ≤ 0.2 a slight change from 0.8 to 0.5 would have been seen, but still. I doubt that a simple test for the x-values of red and grey dots would have been significant. This leaves me wondering if the procedure the authors applied for categorization is meaningful, or if their data is indeed overwhelmingly dominated by noise -- to a degree that it is uninterpretable.

Another example, where I felt that my concern was not addressed adequately was related to the minor comment #3, pointing out to an apparent mismatch in their data with relatively low dependence of experimentally measured DSIs and the position along the tonotopic axis and their model, which shows in contrast a very high dependence of the DSI and the position along the tonotopic axis. Although the authors state in their rebuttal that: "...all our experimental data are coherent with our modeling result. We changed the scale of the plots to emphasize this dependence in the updated figures."

I took the freedom to take the respective panels and to re-plot them according to scale and insist on the privilege to disagree: this does not strike me as coherence between experiment and model. Furthermore, I find it questionable to scale plots such that they match the desired statement instead of facilitating their comparability.

I do appreciate the additional experiments showing that the optogenetic inhibition of PV and SOM cells shows a consistent effect over a wider range of light intensities and additional, valuable experiments potentially helping to clarify a discussion in the field on the impact of different kinds of anesthesia on the ability to observe direction selectivity.

Reviewer #3:

Remarks to the Author:

The authors have done a nice job with their revision. I have no further comments.

-Daniel Polley

Response to Reviewer #2

We are sorry to hear that Reviewer #2 still feels that *“I therefore do not want to block this manuscript, but still must say that I do not feel that major concerns were addressed adequately.”* Although this reviewer admitted in the first round of review that *“their main conclusion that this particular tuning property is calculated at the cortical level is probably right,”* they still showed concern about the lack of description on the local variability in the data that cannot be explained by our network model. Below, please see our responses to individual comments, which explain the source of this local variability and how we address this in the revised text.

“Again, my first major concern was/is: the effect of global biases in the preferred DSI is rather small compared to the large heterogeneity that they report for a given spot along the tonotopic axis. This local heterogeneity represents a much larger fraction of the overall variability in the DSIs than the bias in DSI that is seen along the tonotopic axis. The authors argue in their rebuttal that this heterogeneity is essentially due to trial to trial-noise and can therefore be disregarded. I pointed out to Figure 1d, where they claim to show three representative examples of DSI measurements. Averaged sound responses were calculated to diminish trial-to-trial variability and they obtain impressively clean and nice and gradual direction tuning curves. That these three particular examples have been recorded from different spots along the tonotopic axis was not the point I was making. The point is, that if Figure 1d is true and representative, their assessment of the DSI of a given cell is rather exact and not noise-dominated at all. When the authors then would take their own data serious, as plotted in Figure 1h, or even Figure 1g, they would acknowledge that apparently a lot of real, functional heterogeneity in DSIs exist at the local scale, which, however, is simply ignored by their proposed model:”

Figure A. High-magnification image of the DSI map in our original manuscript, which this reviewer cropped and blew up. We added asterisks to three cells with outlier values.

Zoom in in Figure 1g, left homogeneity in BF, right heterogeneity in DSI The response of the authors to my concern was essentially to apply another data selection criterion such that the analyses better fit the statements they want to make, however, even now, there is still a lot of local DSI heterogeneity. I think when their paper will be read by critical readers, they could reach a higher acceptance by at least adjusting their conclusions and to acknowledge what their data says (much of the DSI heterogeneity is local and not captured by their proposed model / mechanism) and then qualify, that they

are nevertheless focusing on this global bias. This is legitimate. But just ignoring the major effect in their data will leave the reader puzzled.”

We apologize about the confusion regarding this DSI map, since we applied our data selection criterion for quantification figures (e.g. Fig. 1f, 1h), but not for this DSI map—the map that was shown in our first revision was the same as that in our original submission. We did not apply the selection for this example data, reasoning that the visualization of the raw example data would benefit from transparency. However, it seems that this inconsistency has caused a confusion. We now updated our Fig. 1g with the DSI maps using the same selection criterion as the data quantification. We duplicate the plots in the right Figure B, with the blow-up image of the same area that this reviewer pointed to. As you can see, applying the same selection criterion as our quantification preferentially eliminated the outlier cells (cells with asterisks in Figure A). This means that the “local heterogeneity” that this reviewer emphasized is seen mostly in cells with weak near-threshold responses, which are more susceptible to trial-variance, but if we focus on the cells with robust responses, the global effect of our findings is more clear.

This also answers why our representative cells in Figure 1d have *“impressively clean and nice and gradual direction tuning curves.”* Here, we are showcasing representative cells with robust responses with upward, downward, and no direction preference, and they are of course less susceptible to the trial-variance.

Similar selection criteria-dependence of apparent local heterogeneity has been previously reported for the frequency preference of A1 neurons by the Polley lab (Romero et al., *Cereb Cortex*, 2020), and it is very likely that the conflicts between laboratories regarding A1 local heterogeneity is partly due to the mismatch between their data selection criteria. To provide the field with more information regarding this argument, we now included in Supplementary Fig. 4 a direct comparison between data with no selection, middle-threshold, and high-threshold (shown above as Figure C as well). This data shows how local heterogeneity is reduced as we select cells with more robust responses, as is apparent from both R and p values. We now described this observation in the imaging section of Results as:

“We also note a significant local heterogeneity in DSI values, which did not stringently follow the global trend. This heterogeneity became less prominent as we raised the data selection threshold (Supplementary Fig. 4), indicating that cells with less robust responses showed larger variability in DSI. This selection criteria-dependence of local heterogeneity is reminiscent of that in A1 tonotopy³⁶, and thus it might represent a general rule of spatial organization in this area.”

and added a sentence in the modeling section of Results:

“To simplify our analysis, we modeled individual neuronal activity with a dynamic firing rate and organized neurons along a one-dimensional spatial continuum of tonotopy (Fig. 4b). Although this simplification ignores the local heterogeneity in DSI distribution (Fig. 1h), it aims to capture the global trend for the dependence of DSI on the preferred frequency of individual neurons.”

We hope this convinces the reviewer that cells with robust responses adhere better to the global trend. Nevertheless, we are also aware of the remaining heterogeneity which could reflect other direction selectivity mechanisms. Therefore, we also discussed this in Discussion by rewriting a section:

“However, we do not exclude the possibility that our model works in combination with other mechanisms that generate direction selectivity. For example, we observed a relatively low direction selectivity at extremely low (2.5 oct/s) and high (80-160 oct/s) FM rates in both experimental measurements of postsynaptic charges as well as model-based prediction, and these results deviate from the more flat relationship between DSI of neuronal firing and FM rates. We also observed local heterogeneity in DSI especially in cells with weak sweep responses, which cannot be explained by our one-dimensional A1 model. Direction selectivity at these conditions could be partially attributed to other mechanisms, such as delay-and-compare between ON and OFF responses, combination-sensitive supralinear summation, intensity ramp-selective firing (see Supplementary Text), and inheritance from upstream structures (see below).”

“The second major concern was that the sweep stimuli used in the study – at the cortical level – may confound two parameters: pitch and intensity. The author’s

argument in the rebuttal that their observations could not have anything to do with the findings of Deneux et al, since they used stimuli in which only one parameter is changed (namely intensity) is hard to follow for me. I think it is an improvement, that the authors at least mention this possible confound now in the discussion section.”

This reviewer is referring to Deneux et al., *Nat Commun.* (10.1038/ncomms12682), which showed that auditory cortex neurons preferentially respond to intensity ramp-up, as opposed to ramp-down, sounds with fixed spectral content. In our previous revision, we gave rather detailed explanations on why we do not think findings in Deneux et al. explain our data, especially the fact that the intensity ramp-up preference is inconsistent with our data on spectrally-restricted FM sweeps (Fig. 6), which this reviewer may have missed. To make this point more clear, we now include a new section in Supplementary Text titled “Discussion on the contribution of intensity ramp preference” (see below for the full text) that describes in more detail why this mechanism fails to explain our main findings, while at the same time could still influence the local variability of DSI. We chose to include this in Supplementary Text because inclusion of this long explanation in the limited length of text (which we are already exceeding since we have added significant amount of text and data in response to all three reviewers) would confuse, rather than help, the readers.

“Discussion on the contribution of intensity ramp preference

In the mouse auditory system, hearing sensitivity is the highest around 10-30 kHz and decreases towards the edges of the hearing range. Therefore, FM sweeps with a constant sound level will nevertheless provide an input that changes not only on the sound frequency, but also on the intensity that the system perceives. A previous study found that mouse auditory cortex neurons (including A1 and other areas) preferentially respond to intensity ramp-up, as opposed to ramp-down, sounds with fixed spectral content. This raises a question whether the observed FM direction selectivity in A1 neurons is due to the perceived ramps in the sound intensity. We believe that our main findings are not explained by this mechanism for the following reasons:

Although the circuit mechanisms underlying the intensity ramp-up preference are unknown, we can think of two alternative scenarios: 1) intensity ramp-up preference is calculated within each neuron, such that ramp-up of the synaptic input onto each neuron is critical for this computation, or 2) intensity ramp-up preference is calculated in the whole network, such that ramp-up of the overall inputs to the auditory cortex is critical for this computation.

Scenario 1) is obviously against our main findings on 4-octave FM sweeps data. A neuron in low-frequency A1 (e.g. BF = 4 kHz) would experience a ramp-down of synaptic inputs for upward FM sweeps and a ramp-up of synaptic inputs for downward FM sweeps. Therefore, this scenario would predict a downward FM sweep preference in low frequency A1 neurons, which is completely the opposite of our findings in Fig. 1h.

Scenario 2), in contrast, could contribute to the results in Fig. 1h. However, this scenario in turn fails to explain our results on spectrally-restricted FM sweeps (Fig. 6), since this scenario predicts fixed DSI of individual neurons regardless of the FM frequency range. For example, upward FM sweeps of 4→8 kHz and 8→16 kHz both cause an intensity ramp-up of overall inputs to the auditory cortex, thus there should not be a difference in DSI between these two FM stimuli in individual neurons. What we demonstrated in Fig. 6b and 6e is clearly inconsistent with this prediction—we showed that DSI of individual neurons is not a fixed value but rather depends on the relationship between the neuron's BF and the FM frequency range. For example, a neuron with 8 kHz BF would show a downward-preference for 4-8 kHz FM range and an upward-preference for 8-16 kHz FM range.

Together, in either scenario, intensity ramp-up preference fails to account for our results on FM sweep direction selectivity. However, we do not exclude the possibility that this mechanism, together with other direction selectivity mechanisms, such as delay-and-compare between ON and OFF responses, combination-sensitive supralinear summation, and inheritance from upstream structures, contributes to the local heterogeneity in DSI that are not captured by the global trend that our model explains.”

“Another major point was the data shown in Figure 3g. Again, here, the argument of noisiness is being used to try to address my concern: I found the plot difficult to read, as there are apparently many cells that have been classified as sweep selective, but not direction selective (grey), that have surprisingly an DSI of 1 (grey arrows). Also, there are several cells that have been classified as both, sweep and direction selective (red), that barely show any direction selectivity as quantified with DSI (red arrows).

Sure, noisiness can lead to a surprisingly high DSI, that turns out not to be significant (caused e.g. by a single major response) and if suddenly the trial to trial variability in a given cell apparently is very low, one could observe a small DSI, that is nevertheless significant. But overall, if some statistics is applied to achieve a meaningful categorization into direction selective and non-selective cells, one would nevertheless expect some separation of the data according to the categories. Here, I just consider the x-values of the puncta and their distribution along the x-axis, describing the DSIs under basal, control conditions. If grey dots should represent non-direction selective cells, I would expect a relative enrichment of 'grey' cells to the left, and if the red dots should represent direction selective cells, they should be relatively enriched for higher DSI values. For the x-values smaller than 0.4, I counted 9 grey / 15 red (0.6) and for x-values larger than 0.4, I counted 4 grey / 6 red (0.66), essentially a similar ratio. Admittedly, if one would have made this calculation with a threshold of DSIs $\ll 0.2$ a slight change from 0.8 to 0.5 would have been seen, but still. I doubt that a simple test for the x-values of red and grey dots would have been significant. This leaves me

wondering if the procedure the authors applied for categorization is meaningful, or if their data is indeed overwhelmingly dominated by noise -- to a degree that it is uninterpretable.”

We believe that our discussion above has already answered this reviewer’s concern. Neurons with near-threshold responses (which are more likely to be non-significant) often show extreme DSI values due to the nature of DSI calculation, and that is why testing for statistical significance is meaningful. Nevertheless, to address this reviewer’s concern and avoid potential confusion, we updated our Figures 3 and S7 with the data without classification.

As we stated in our original manuscript, even without classification, the conclusions from our experiments including all the data points stay the same and highly significant (SOM-eNpHR3: $p = 0.0006$; PV-eNpHR3: $p = 0.667$).

“Another example, where I felt that my concern was not addressed adequately was related to the minor comment #3, pointing out to an apparent mismatch in their data with relatively low dependence of experimentally measured DSIs and the position along the tonotopic axis and their model, which shows in contrast a very high dependence of the DSI and the position along the tonotopic axis. Although the authors state in their rebuttal that: “...all our experimental data are coherent with our modeling result. We changed the scale of the plots to emphasize this dependence in the updated figures.”

I took the freedom to take the respective panels and to re-plot them according to scale and insist on the privilege to disagree: this does not strike me as coherence between experiment and model. Furthermore, I find it questionable to scale plots such that they match the desired statement instead of facilitating their comparability.”

In our original manuscript, we have explicitly pointed out the difference between the DSI of firing and DSI of synaptic currents (which is closer to our model) at extremely low and high FM rates and suggested the contributions of other mechanisms by stating that “However, we do not exclude the possibility that our model works in combination with other mechanisms that generate direction selectivity. For example, we observed a relatively low direction selectivity of postsynaptic charges at extremely low (2.5 oct/s) and high (80-160 oct/s) FM rates. Direction selectivity at these conditions could be partially attributed to other mechanisms, such as delay-and-compare between ON and OFF responses, combination-sensitive supralinear summation, and inheritance from upstream structures.” (In the previous revision,

we also included intensity-ramp mechanism in this sentence, following the comment of this reviewer.) We believe this sentence is exactly what this reviewer is requesting, but maybe it was not explicit enough. We rewrote this section to more explicitly point to the difference between the model and the experiments, and described the potential reasons for the mismatch between the model and our experimental data, as we already described in our response to this reviewer's first comment:

"However, we do not exclude the possibility that our model works in combination with other mechanisms that generate direction selectivity. For example, we observed a relatively low direction selectivity at extremely low (2.5 oct/s) and high (80-160 oct/s) FM rates in both experimental measurements of postsynaptic charges as well as model-based prediction, and these results deviate from the more flat relationship between DSI of neuronal firing and FM rates. We also observed local heterogeneity in DSI especially in cells with weak sweep responses, which cannot be explained by our one-dimensional A1 model. Direction selectivity at these conditions could be partially attributed to other mechanisms, such as delay-and-compare between ON and OFF responses, combination-sensitive supralinear summation, intensity ramp-selective firing (see Supplementary Text), and inheritance from upstream structures (see below)."

Furthermore, we followed this reviewer's advice and matched the y axis for all the absolute DSI vs absolute rate plots, including Figures 1f, 2e, 4g, 6d, and S4c. The important point that we want to emphasize by these figures is not affected by these changes—the main claim of these figures is that A1 neurons are direction-selective to ethologically relevant slow FM sweeps, in contrast to the previous paper by Zhang et al. (*Nature* 2003) that showed no direction selectivity below 30 oct/s.

"I do appreciate the additional experiments showing that the optogenetic inhibition of PV and SOM cells shows a consistent effect over a wider range of light intensities and additional, valuable experiments potentially helping to clarify a discussion in the field on the impact of different kinds of anesthesia on the ability to observe direction selectivity."

We are glad to hear that this reviewer appreciates these new experiments and analyses that we added for our previous revision. We agree that these additions made our manuscript more solid and informative to the field.